# Regret of Queueing Bandits

**Subhashini Krishnasamy**
University of Texas at Austin

**Rajat Sen**
University of Texas at Austin

**Ramesh Johari**
Stanford University

**Sanjay Shakkottai**
University of Texas at Austin

## Abstract

We consider a variant of the multiarmed bandit problem where jobs *queue* for service, and service rates of different servers may be unknown. We study algorithms that minimize *queue-regret*: the (expected) difference between the queue-lengths obtained by the algorithm, and those obtained by a "genie"-aided matching algorithm that knows exact service rates. A naive view of this problem would suggest that queue-regret should grow logarithmically: since queue-regret cannot be larger than classical regret, results for the standard MAB problem give algorithms that ensure queue-regret increases no more than logarithmically in time. Our paper shows surprisingly more complex behavior. In particular, the naive intuition is correct as long as the bandit algorithm's queues have relatively long regenerative cycles: in this case queue-regret is similar to cumulative regret, and scales (essentially) logarithmically. However, we show that this "early stage" of the queueing bandit eventually gives way to a "late stage", where the optimal queue-regret scaling is $O(1/t)$. We demonstrate an algorithm that (order-wise) achieves this asymptotic queue-regret, and also exhibits close to optimal switching time from the early stage to the late stage.

## 1 Introduction

Stochastic multi-armed bandits (MAB) have a rich history in sequential decision making [1, 2, 3]. In its simplest form, a collection of $K$ arms are present, each having a binary reward (Bernoulli random variable over $\{0, 1\}$) with an unknown success probability[1] (and different across arms). At each (discrete) time, a single arm is chosen by the bandit algorithm, and a (binary-valued) reward is accrued. The MAB problem is to determine which arm to choose at each time in order to minimize the cumulative expected regret, namely, the cumulative loss of reward when compared to a genie that has knowledge of the arm success probabilities.

In this paper, we consider the variant of this problem motivated by *queueing* applications. Formally, suppose that arms are pulled upon arrivals of *jobs*; each arm is now a *server* that can serve the arriving job. In this model, the stochastic reward described above is equivalent to *service*. In other words, if the arm (server) that is chosen results in positive reward, the job is successfully completed and departs the system. However, this basic model fails to capture an essential feature of service in many settings: in a queueing system, *jobs wait until they complete service*. Such systems are *stateful*: when the chosen arm results in zero reward, the job being served remains in the queue, and over time the model must track the remaining jobs waiting to be served. The difference between the cumulative number of arrivals and departures, or the *queue length*, is the most common measure of the quality of the service strategy being employed.

Queueing is employed in modeling a vast range of service systems, including supply and demand in online platforms (e.g., Uber, Lyft, Airbnb, Upwork, etc.); order flow in financial markets (e.g., limit order books); packet flow in communication networks; and supply chains. In all of these systems, queueing is an essential part of the model: e.g., in online platforms, the available supply (e.g. available drivers in Uber or Lyft, or available rentals in Airbnb) queues until it is "served" by arriving demand (ride requests in Uber or Lyft, booking requests in Airbnb). Since MAB models are a natural way to capture learning in this entire range of systems, incorporating queueing behavior into the MAB model is an essential challenge.

This problem clearly has the explore-exploit tradeoff inherent in the standard MAB problem: since the success probabilities across different servers are unknown, there is a tradeoff between learning (*exploring*) the different servers and (*exploiting*) the most promising server from past observations. We refer to this problem as the *queueing bandit*. Since the queue length is simply the difference between the cumulative number arrivals and departures (cumulative actual reward; here reward is 1 if job is served), the natural notion of regret here is to compare the expected queue length under a bandit algorithm with the corresponding one under a genie policy (with identical arrivals) that however always chooses the arm with the highest expected reward.

**Queueing System:** To capture this trade-off, we consider a discrete-time queueing system with a single queue and $K$ servers. Arrivals to the queue and service offered by the links are according to product Bernoulli distribution and i.i.d. across time slots. Statistical parameters corresponding to the service distributions are considered unknown. In any time slot, the queue can be served by at most one server and the problem is to schedule a server in every time slot. The service is *pre-emptive* and a job returns to the queue if not served. There is at least one server that has a service rate higher than the arrival rate, which ensures that the "genie" policy is stable.

Let $Q(t)$ be the queue length at time $t$ under a given bandit algorithm, and let $Q^*(t)$ be the corresponding queue length under the "genie" policy that always schedules the optimal server (i.e. always plays the arm with the highest mean). We define the *queue-regret* as the difference in expected queue lengths for the two policies. That is, the regret is given by:

$$\Psi(t) := \mathbb{E}\left[Q(t) - Q^*(t)\right]. \tag{1}$$

Here $\Psi(t)$ has the interpretation of the traditional MAB regret with caveat that rewards are accumulated only if there is a job that can benefit from this reward. We refer to $\Psi(t)$ as the *queue-regret*; formally, our goal is to develop bandit algorithms that minimize the queue-regret at a finite time $t$.

To develop some intuition, we compare this to the standard stochastic MAB problem. For the standard problem, well-known algorithms such as UCB, KL-UCB, and Thompson sampling achieve a cumulative regret of $O((K-1)\log t)$ at time $t$ [4, 5, 6], and this result is essentially tight [7]. In the queueing bandit, we can obtain a simple bound on the queue-regret by noting that it cannot be any higher than the traditional regret (where a reward is accrued at each time whether a job is present or not). This leads to an upper bound of $O((K-1)\log t)$ for the queue regret.

However, this upper bound does not tell the whole story for the queueing bandit: we show that there are two "stages" to the queueing bandit. In the *early* stage, the bandit algorithm is unable to even stabilize the queue – i.e. on average, the queue length increases over time and is continuously backlogged; therefore the queue-regret grows with time, similar to the cumulative regret. Once the algorithm is able to stabilize the queue—the *late* stage—then a dramatic shift occurs in the behavior of the queue regret. A stochastically stable queue goes through **regenerative cycles** – a random cyclical behavior where queues build-up over time, then empty, and the cycle repeats. The associated recurring"zero-queue-length" epochs means that sample-path queue-regret essentially "resets" at (stochastically) regular intervals; i.e., the sample-path queue-regret becomes non-positive at these time instants. Thus the queue-regret should fall over time, as the algorithm learns.

Our main results provide lower bounds on queue-regret for both the early and late stages, as well as algorithms that essentially match these lower bounds. We first describe the late stage, and then describe the early stage for a heavily loaded system.

**1. The late stage**. We first consider what happens to the queue regret as $t \to \infty$. As noted above, a reasonable intuition for this regime comes from considering a standard bandit algorithm, but where the sample-path queue-regret "resets" at time points of regeneration.[2] In this case, the queue-regret is

approximately a (discrete) *derivative* of the cumulative regret. Since the optimal cumulative regret scales like $\log t$, asymptotically the optimal queue-regret should scale like $1/t$. Indeed, we show that the queue-regret for $\alpha$-consistent policies is at least $C/t$ infinitely often, where $C$ is a constant independent of $t$. Further, we introduce an algorithm called Q-ThS for the queueing bandit (a variant of Thompson sampling with explicit structured exploration), and show an asymptotic regret upper bound of $O\left(\text{poly}(\log t)/t\right)$ for Q-ThS, thus matching the lower bound up to poly-logarithmic factors in $t$. Q-ThS exploits *structured exploration*: we exploit the fact that the queue regenerates regularly to explore more systematically and aggressively.

**2. The early stage**. The preceding discussion might suggest that an algorithm that explores aggressively would dominate any algorithm that balances exploration and exploitation. However, an overly aggressive exploration policy will preclude the queueing system from ever stabilizing, which is *necessary* to induce the regenerative cycles that lead the system to the late stage. To even enter the late stage, therefore, we need an algorithm that exploits enough to actually stabilize the queue (i.e. choose good arms sufficiently often so that the mean service rate exceeds the expected arrival rate).

We refer to the early stage of the system, as noted above, as the period before the algorithm has learned to stabilize the queues. For a *heavily loaded system, where the arrival rate approaches the service rate of the optimal server,* we show a lower bound of $\Omega(\log t/\log\log t)$ on the queue-regret in the early stage. Thus up to a $\log\log t$ factor, the early stage regret behaves similarly to the cumulative regret (which scales like $\log t$). The heavily loaded regime is a natural asymptotic regime in which to study queueing systems, and has been extensively employed in the literature; see, e.g., [9, 10] for surveys.

Perhaps more importantly, our analysis shows that the time to switch from the early stage to the late stage scales at least as $t = \Omega(K/\epsilon)$, where $\epsilon$ is the gap between the arrival rate and the service rate of the optimal server; thus $\epsilon \to 0$ in the heavy-load setting. In particular, we show that the early stage lower bound of $\Omega(\log t/\log\log t)$ is valid up to $t = O(K/\epsilon)$; on the other hand, we also show that, in the heavy-load limit, depending on the relative scaling between $K$ and $\epsilon$, the regret of Q-ThS scales like $O\left(\text{poly}(\log t)/\epsilon^2 t\right)$ for times that are arbitrarily close to $\Omega(K/\epsilon)$. In other words, Q-ThS is nearly optimal in the time it takes to "switch" from the early stage to the late stage.

Our results constitute the first insight into the behavior of regret in this queueing setting; as emphasized, it is quite different than that seen for minimization of cumulative regret in the standard MAB problem. The preceding discussion highlights why minimization of queue-regret presents a subtle learning problem. On one hand, if the queue has been stabilized, the presence of regenerative cycles allows us to establish that queue regret must eventually decay to zero at rate $1/t$ under an optimal algorithm (the late stage). On the other hand, to actually have regenerative cycles in the first place, a learning algorithm needs to exploit enough to actually stabilize the queue (the early stage). Our analysis not only characterizes regret in both regimes, but also essentially exactly characterizes the transition point between the two regimes. In this way the queueing bandit is a remarkable new example of the tradeoff between exploration and exploitation.

## 2   Related work

**MAB algorithms**. Stochastic MAB models have been widely used in the past as a paradigm for various sequential decision making problems in industrial manufacturing, communication networks, clinical trials, online advertising and webpage optimization, and other domains requiring resource allocation and scheduling; see, e.g., [1, 2, 3]. The MAB problem has been studied in two variants, based on different notions of optimality. One considers mean accumulated loss of rewards, often called *regret*, as compared to a genie policy that always chooses the best arm. Most effort in this direction is focused on getting the best regret bounds possible at any *finite time* in addition to designing computationally feasible algorithms [3]. The other line of research models the bandit problem as a Markov decision process (MDP), with the goal of optimizing *infinite horizon* discounted or average reward. The aim is to characterize the structure of the optimal policy [2]. Since these policies deal with optimality with respect to infinite horizon costs, unlike the former body of research, they give steady-state and not finite-time guarantees. Our work uses the regret minimization framework to study the queueing bandit problem.

**Bandits for queues**. There is body of literature on the application of bandit models to queueing and scheduling systems [2, 11, 12, 13, 14, 15, 16, 17]. These queueing studies focus on infinite-horizon

costs (i.e., statistically steady-state behavior, where the focus typically is on conditions for optimality of index policies); further, the models do not typically consider user-dependent server statistics. Our focus here is different: algorithms and analysis to optimize finite time regret.

## 3  Problem Setting

We consider a discrete-time queueing system with a single queue and $K$ servers. The servers are indexed by $k = 1, \ldots, K$. Arrivals to the queue and service offered by the links are according to product Bernoulli distribution and i.i.d. across time slots. The mean arrival rate is given by $\lambda$ and the mean service rates by the vector $\boldsymbol{\mu} = [\mu_k]_{k \in [K]}$, with $\lambda < \max_{k \in [K]} \mu_k$. In any time slot, the queue can be served by at most one server and the problem is to schedule a server in every time slot. The scheduling decision at any time $t$ is based on past observations corresponding to the services obtained from the scheduled servers until time $t - 1$. Statistical parameters corresponding to the service distributions are considered unknown. The queueing system evolution can be described as follows. Let $\kappa(t)$ denote the server that is scheduled at time $t$. Also, let $R_k(t) \in \{0, 1\}$ be the service offered by server $k$ and $S(t)$ denote the service offered by server $\kappa(t)$ at time $t$, i.e., $S(t) = R_{\kappa(t)}(t)$. If $A(t)$ is the number of arrivals at time $t$, then the queue-length at time $t$ is given by: $Q(t) = (Q(t - 1) + A(t) - S(t))^+$.

Our goal in this paper is to focus attention on how queueing behavior impacts regret minimization in bandit algorithms. We evaluate the performance of scheduling policies against the policy that schedules the (unique) optimal server in every time slot, i.e., the server $k^* := \arg \max_{k \in [K]} \mu_k$ with the maximum mean rate $\mu^* := \max_{k \in [K]} \mu_k$. Let $Q(t)$ be the queue-length vector at time $t$ under our specified algorithm, and let $Q^*(t)$ be the corresponding vector under the optimal policy. We define *regret* as the difference in mean queue-lengths for the two policies. That is, the regret is given by: $\Psi(t) := \mathbb{E}\left[Q(t) - Q^*(t)\right]$. We use the terms *queue-regret* or simply *regret* to refer to $\Psi(t)$.

Throughout, when we evaluate queue-regret, we do so under the assumption that the queueing system starts in the steady state distribution of the system induced by the optimal policy, as follows.

**Assumption 1** (Initial State). *Both $Q(0)$ and $Q^*(0)$ have the same initial state distribution, and this is chosen to be the stationary distribution of $Q^*(t)$; this distribution is denoted $\pi_{(\lambda, \mu^*)}$.*

## 4  The Late Stage

We analyze the performance of a scheduling algorithm with respect to queue-regret as a function of time and system parameters like: *(a)* the load on the system $\epsilon := (\mu^* - \lambda)$, and *(b)* the minimum difference between the rates of the best and the next best servers $\Delta := \mu^* - \max_{k \neq k^*} \mu_k$.

As a preview of the theoretical results, Figure 1 shows the evolution of queue-regret with time in a system with 5 servers under a scheduling policy inspired by Thompson Sampling. Exact details of the scheduling algorithm can be found in Section 4.2. It is observed that the regret goes through a phase transition. In the initial stage, when the algorithm has not estimated the service rates well enough to stabilize the queue, the regret grows poly-logarithmically similar to the classical MAB setting. After a critical point when the algorithm has learned the system parameters well enough to stabilize the queue, the queue-length goes through regenerative cycles as the queue

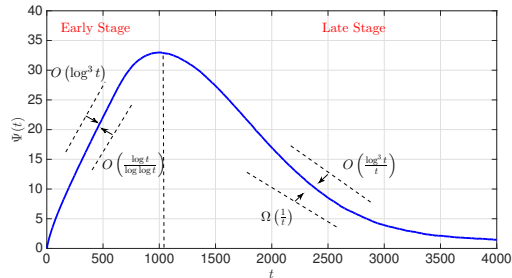

Figure 1: Queue-regret $\Psi(t)$ under Q-ThS in a system with $K = 5$, $\epsilon = 0.1$ and $\Delta = 0.17$

become empty. In other-words, instead of the queue length being continuously backlogged, the queuing system has a stochastic cyclical behavior where the queue builds up, becomes empty, and this cycle recurs. Thus at the beginning of every regenerative cycle, there is no accumulation of past errors and the sample-path queue-regret is at most zero. As the algorithm estimates the parameters better with time, the length of the regenerative cycles decreases and the queue-regret decays to zero.

**Notation**: For the results in Section 4, the notation $f(t) = O\left(g(K, \epsilon, t)\right)$ for all $t \in h(K, \epsilon)$ (here, $h(K, \epsilon)$ is an interval that depends on $K, \epsilon$) implies that there exist constants $C$ and $t_0$ independent of $K$ and $\epsilon$ such that $f(t) \leq Cg(K, \epsilon, t)$ for all $t \in (t_0, \infty) \cap h(K, \epsilon)$.

## 4.1 An Asymptotic Lower Bound

We establish an asymptotic lower bound on regret for the class of $\alpha$-consistent policies; this class for the queueing bandit is a generalization of the $\alpha$-consistent class used in the literature for the traditional stochastic MAB problem [7, 18, 19]. The precise definition is given below ($\mathbb{1}\{\cdot\}$ below is the indicator function).

**Definition 1.** *A scheduling policy is said to be $\alpha$-consistent (for some $\alpha \in (0,1)$) if given any problem instance, specified by $(\lambda, \boldsymbol{\mu})$, $\mathbb{E}\left[\sum_{s=1}^{t} \mathbb{1}\{\kappa(s) = k\}\right] = O(t^{\alpha})$ for all $k \neq k^*$.*

Theorem 1 below gives an asymptotic lower bound on the average queue-regret and per-queue regret for an arbitrary $\alpha$-consistent policy.

**Theorem 1.** *For any problem instance $(\lambda, \boldsymbol{\mu})$ and any $\alpha$-consistent policy, the regret $\Psi(t)$ satisfies*

$$\Psi(t) \geq \left(\frac{\lambda}{4} D(\boldsymbol{\mu})(1 - \alpha)(K - 1)\right) \frac{1}{t}$$

*for infinitely many $t$, where*

$$D(\boldsymbol{\mu}) = \frac{\Delta}{\text{KL}\left(\mu_{min}, \frac{\mu^*+1}{2}\right)}. \tag{2}$$

*Outline for theorem 1.* The proof of the lower bound consists of three main steps. First, in lemma 21, we show that the regret at any time-slot is lower bounded by the probability of a sub-optimal schedule in that time-slot (up to a constant factor that is dependent on the problem instance). The key idea in this lemma is to show the equivalence of any two systems with the same marginal service distributions under bandit feedback. This is achieved through a carefully constructed coupling argument that maps the original system with independent service across links to another system with service process that is dependent across links but with the same marginal distribution.

As a second step, the lower bound on the regret in terms of the probability of a sub-optimal schedule enables us to obtain a lower bound on the cumulative queue-regret in terms of the number of sub-optimal schedules. We then use a lower bound on the number of sub-optimal schedules for $\alpha$-consistent policies (lemma 19 and corollary 20) to obtain a lower bound on the cumulative regret. In the final step, we use the lower bound on the cumulative queue-regret to obtain an *infinitely often* lower bound on the queue-regret. $\qquad \square$

## 4.2 Achieving the Asymptotic Bound

We next focus on algorithms that can (up to a poly log factor) achieve a scaling of $O\left(1/t\right)$. A key challenge in showing this is that we will need high probability bounds on the number of times the correct arm is scheduled, and these bounds to hold over the late-stage regenerative cycles of the queue. Recall that these regenerative cycles are random time intervals with $\Theta(1)$ expected length for the optimal policy, and whose lengths are correlated with the bandit algorithm decisions (the queue length evolution is dependent on the past history of bandit arm schedules). To address this, we propose a slightly modified version of the Thompson Sampling algorithm. The algorithm, which we call Q-ThS, has an explicit structured exploration component similar to $\epsilon$-greedy algorithms. This structured exploration provides sufficiently good estimates for all arms (including sub-optimal ones) in the late stage.

We describe the algorithm we employ in detail. Let $T_k(t)$ be the number of times server $k$ is assigned in the first $t$ time-slots and $\hat{\boldsymbol{\mu}}(t)$ be the empirical mean of service rates at time-slot $t$ from past observations (until $t - 1$). At time-slot $t$, Q-ThS decides to *explore* with probability $\min\{1, 3K \log^2 t/t\}$, otherwise it *exploits*. When exploring, it chooses a server uniformly at random. The chosen exploration rate ensures that we are able to obtain concentration results for the number

of times any link is sampled.[3] When exploiting, for each $k \in [K]$, we pick a sample $\hat{\theta}_k(t)$ of distribution $\text{Beta}\left(\hat{\mu}_k(t) T_k(t-1) + 1, (1 - \hat{\mu}_k(t)) T_k(t-1) + 1\right)$, and schedule the arm with the largest sample (the standard Thompson sampling for Bernoulli arms [20]). Details of the algorithm are given in Algorithm 1 in the Appendix.

We now show that, for a given problem instance $(\lambda, \boldsymbol{\mu})$ (and therefore fixed $\epsilon$), the regret under Q-ThS scales as $O\left(\text{poly}(\log t)/t\right)$. We state the most general form of the asymptotic upper bound in theorem 2. A slightly weaker version of the result is given in corollary 3. This corollary is useful to understand the dependence of the upper bound on the load $\epsilon$ and the number of servers $K$.

**Notation** : For the following results, the notation $f(t) = O\left(g(K, \epsilon, t)\right)$ for all $t \in h(K, \epsilon)$ (here, $h(K, \epsilon)$ is an interval that depends on $K, \epsilon$) implies that there exist constants $C$ and $t_0$ independent of $K$ and $\epsilon$ such that $f(t) \leq Cg(K, \epsilon, t)$ for all $t \in (t_0, \infty) \cap h(K, \epsilon)$.

**Theorem 2.** *Consider any problem instance $(\lambda, \boldsymbol{\mu})$. Let $w(t) = \exp\left(\left(\frac{2\log t}{\Delta}\right)^{2/3}\right)$, $v'(t) = \frac{6K}{\epsilon} w(t)$ and $v(t) = \frac{24}{\epsilon^2}\log t + \frac{60K}{\epsilon}\frac{v'(t)\log^2 t}{t}$. Then, under Q-ThS the regret $\Psi(t)$, satisfies*

$$\Psi(t) = O\left(\frac{Kv(t)\log^2 t}{t}\right)$$

*for all $t$ such that $\frac{w(t)}{\log t} \geq \frac{2}{\epsilon}$, $t \geq \exp\left(6/\Delta^2\right)$ and $v(t) + v'(t) \leq t/2$.*

**Corollary 3.** *Let $w(t)$ be as defined in Theorem 2. Then,*

$$\Psi(t) = O\left(K\frac{\log^3 t}{\epsilon^2 t}\right)$$

*for all $t$ such that $\frac{w(t)}{\log t} \geq \frac{2}{\epsilon}$, $\frac{t}{w(t)} \geq \max\left\{\frac{24K}{\epsilon}, 15K^2\log t\right\}$, $t \geq \exp\left(6/\Delta^2\right)$ and $\frac{t}{\log t} \geq \frac{198}{\epsilon^2}$.*

*Outline for Theorem 2.* As mentioned earlier, the central idea in the proof is that the sample-path queue-regret is at most zero at the beginning of regenerative cycles, i.e., instants at which the queue becomes empty. The proof consists of two main parts – one which gives a high probability result on the number of sub-optimal schedules in the exploit phase in the late stage, and the other which shows that at any time, the beginning of the current regenerative cycle is not very far in time.

The former part is proved in lemma 9, where we make use of the structured exploration component of Q-ThS to show that all the links, including the sub-optimal ones, are sampled a sufficiently large number of times to give a good estimate of the link rates. This in turn ensures that the algorithm schedules the correct link in the exploit phase in the late stages with high probability.

For the latter part, we prove a high probability bound on the last time instant when the queue was zero (which is the beginning of the current regenerative cycle) in lemma 15. Here, we make use of a recursive argument to obtain a tight bound. More specifically, we first use a coarse high probability upper bound on the queue-length (lemma 11) to get a first cut bound on the beginning of the regenerative cycle (lemma 12). This bound on the regenerative cycle-length is then recursively used to obtain tighter bounds on the queue-length, and in turn, the start of the current regenerative cycle (lemmas 14 and 15 respectively).

The proof of the theorem proceeds by combining the two parts above to show that the main contribution to the queue-regret comes from the structured exploration component in the current regenerative cycle, which gives the stated result. □

## 5 The Early Stage in the Heavily Loaded Regime

In order to study the performance of $\alpha$-consistent policies in the early stage, we consider the *heavily loaded* system, where the arrival rate $\lambda$ is close to the optimal service rate $\mu^*$, i.e., $\epsilon = \mu^* - \lambda \to 0$. This is a well studied asymptotic in which to study queueing systems, as this regime leads to

fundamental insight into the structure of queueing systems. See, e.g., [9, 10] for extensive surveys. Analyzing queue-regret in the early stage in the heavily loaded regime has the effect that the the optimal server is the only one that stabilizes the queue. As a result, in the heavily loaded regime, effective learning and scheduling of the optimal server play a crucial role in determining the transition point from the early stage to the late stage. For this reason the heavily loaded regime reveals the behavior of regret in the early stage.

**Notation:** For all the results in this section, the notation $f(t) = O(g(K, \epsilon, t))$ for all $t \in h(K, \epsilon)$ ($h(K, \epsilon)$ is an interval that depends on $K, \epsilon$) implies that there exist numbers $C$ and $\epsilon_0$ that depend on $\Delta$ such that for all $\epsilon \geq \epsilon_0$, $f(t) \leq Cg(K, \epsilon, t)$ for all $t \in h(K, \epsilon)$.

Theorem 4 gives a lower bound on the regret in the heavily loaded regime, roughly in the time interval $\left(K^{1/(1-\alpha)}, O(K/\epsilon)\right)$ for any $\alpha$-consistent policy.

**Theorem 4.** *Given any problem instance $(\lambda, \boldsymbol{\mu})$, and for any $\alpha$-consistent policy and $\gamma > \frac{1}{1-\alpha}$, the regret $\Psi(t)$ satisfies*

$$\Psi(t) \geq \frac{D(\boldsymbol{\mu})}{2}(K-1)\frac{\log t}{\log \log t}$$

*for $t \in \left[\max\{C_1 K^\gamma, \tau\}, (K-1)\frac{D(\boldsymbol{\mu})}{2\epsilon}\right]$ where $D(\boldsymbol{\mu})$ is given by equation 2, and $\tau$ and $C_1$ are constants that depend on $\alpha$, $\gamma$ and the policy.*

*Outline for Theorem 4.* The crucial idea in the proof is to show a lower bound on the queue-regret in terms of the number of sub-optimal schedules (Lemma 22). As in Theorem 1, we then use a lower bound on the number of sub-optimal schedules for $\alpha$-consistent policies (given by Corollary 20) to obtain a lower bound on the queue-regret. □

Theorem 4 shows that, for any $\alpha$-consistent policy, it takes at least $\Omega(K/\epsilon)$ time for the queue-regret to transition from the early stage to the late stage. In this region, the scaling $O(\log t/\log \log t)$ reflects the fact that queue-regret is dominated by the cumulative regret growing like $O(\log t)$. A reasonable question then arises: after time $\Omega(K/\epsilon)$, should we expect the regret to transition into the late stage regime analyzed in the preceding section?

We answer this question by studying when Q-ThS achieves its late-stage regret scaling of $O(\text{poly}(\log t)/\epsilon^2 t)$ scaling; as we will see, in an appropriate sense, Q-ThS is close to optimal in its transition from early stage to late stage, when compared to the bound discovered in Theorem 4. Formally, we have Corollary 5, which is an analog to Corollary 3 under the heavily loaded regime.

**Corollary 5.** *For any problem instance $(\lambda, \boldsymbol{\mu})$, any $\gamma \in (0, 1)$ and $\delta \in (0, \min(\gamma, 1 - \gamma))$, the regret under Q-ThS satisfies*

$$\Psi(t) = O\left(\frac{K \log^3 t}{\epsilon^2 t}\right)$$

*$\forall t \geq C_2 \max\left\{\left(\frac{1}{\epsilon}\right)^{\frac{1}{\gamma-\delta}}, \left(\frac{K}{\epsilon}\right)^{\frac{1}{1-\gamma}}, (K^2)^{\frac{1}{1-\gamma-\delta}}, \left(\frac{1}{\epsilon^2}\right)^{\frac{1}{1-\delta}}\right\}$, where $C_2$ is a constant independent of $\epsilon$ (but depends on $\Delta$, $\gamma$ and $\delta$).*

By combining the result in Corollary 5 with Theorem 4, we can infer that in the heavily loaded regime, the time taken by Q-ThS to achieve $O(\text{poly}(\log t)/\epsilon^2 t)$ scaling is, in some sense, order-wise close to the optimal in the $\alpha$-consistent class. Specifically, for any $\beta \in (0, 1)$, there exists a scaling of $K$ with $\epsilon$ such that the queue-regret under Q-ThS scales as $O(\text{poly}(\log t)/\epsilon^2 t)$ for all $t > (K/\epsilon)^\beta$ while the regret under any $\alpha$-consistent policy scales as $\Omega(K \log t/\log \log t)$ for $t < K/\epsilon$.

We conclude by noting that while the transition point from the early stage to the late stage for Q-ThS is near optimal in the heavily loaded regime, it does not yield optimal regret performance in the early stage in general. In particular, recall that at any time $t$, the structured exploration component in Q-ThS is invoked with probability $3K \log^2 t/t$. As a result, we see that, in the early stage, queue-regret under Q-ThS could be a $\log^2 t$-factor worse than the $\Omega(\log t/\log \log t)$ lower bound shown in Theorem 4 for the $\alpha$-consistent class. This intuition can be formalized: it is straightforward to show an upper bound of $2K \log^3 t$ for any $t > \max\{C_3, U\}$, where $C_3$ is a constant that depends on $\Delta$ but is independent of $K$ and $\epsilon$; we omit the details.

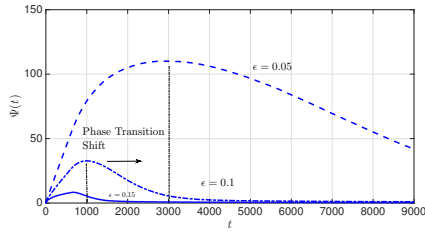
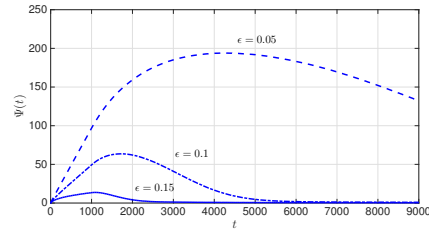

(a) Queue-Regret under Q-ThS for a system with 5 servers with $\epsilon \in \{0.05, 0.1, 0.15\}$

(b) Queue-Regret under Q-ThS for a a system with 7 servers with $\epsilon \in \{0.05, 0.1, 0.15\}$

Figure 2: Variation of Queue-regret $\Psi(t)$ with $K$ and $\epsilon$ under Q-Ths. The phase-transition point shifts towards the right as $\epsilon$ decreases. The efficiency of learning decreases with increase in the size of the system.

# 6 Simulation Results

In this section we present simulation results of various queueing bandit systems with $K$ servers. These results corroborate our theoretical analysis in Sections 4 and 5. In particular a phase transition from unstable to stable behavior can be observed in all our simulations, as predicted by our analysis. In the remainder of the section we demonstrate the performance of Algorithm 1 under variations of system parameters like the traffic ($\epsilon$), the gap between the optimal and the suboptimal servers ($\Delta$), and the size of the system ($K$). We also compare the performance of our algorithm with versions of UCB-1 [4] and Thompson Sampling [20] without structured exploration (Figure 3 in the appendix).

**Variation with $\epsilon$ and** $K$. In Figure 2 we see the evolution of $\Psi(t)$ in systems of size 5 and 7 . It can be observed that the regret decays faster in the smaller system, which is predicted by Theorem 2 in the late stage and Corollary 5 in the early stage. The performance of the system under different traffic settings can be observed in Figure 2. It is evident that the regret of the queueing system grows with decreasing $\epsilon$. This is in agreement with our analytical results (Corollaries 3 and 5). In Figure 2 we can observe that the time at which the phase transition occurs shifts towards the right with decreasing $\epsilon$ which is predicted by Corollaries 3 and 5.

# 7 Discussion and Conclusion

This paper provides the first regret analysis of the queueing bandit problem, including a characterization of regret in both early and late stages, together with analysis of the switching time; and an algorithm (Q-ThS) that is asymptotically optimal (to within poly-logarithmic factors) and also essentially exhibits the correct switching behavior between early and late stages. There remain substantial open directions for future work.

*First*, is there a single algorithm that gives optimal performance in *both* early and late stages, as well as the optimal switching time between early and late stages? The price paid for structured exploration by Q-ThS is an inflation of regret in the early stage. An important open question is to find a single, adaptive algorithm that gives good performance over all time. As we note in the appendix, classic (unstructured) Thompson sampling is an intriguing candidate from this perspective.

*Second* the most significant technical hurdle in finding a single optimal algorithm is the difficulty of establishing concentration results for the number of suboptimal arm pulls within a regenerative cycle whose length is dependent on the bandit strategy. Such concentration results would be needed in two different limits: first, as the start time of the regenerative cycle approaches infinity (for the asymptotic analysis of late stage regret); and second, as the load of the system increases (for the analysis of early stage regret in the heavily loaded regime). Any progress on the open directions described above would likely require substantial progress on these technical questions as well.

**Acknowledgement:** This work is partially supported by NSF Grants CNS-1161868, CNS-1343383, CNS-1320175, ARO grants W911NF-16-1-0377, W911NF-15-1-0227, W911NF-14-1-0387 and the US DoT supported D-STOP Tier 1 University Transportation Center.

## Footnotes

[1]Here, the success probability of an arm is the probability that the reward equals '1'.

[2]This is inexact since the optimal queueing system and bandit queueing system may not regenerate at the same time point; but the intuition holds.

[3]The exploration rate could scale like $\log t/t$ if we knew $\Delta$ in advance; however, without this knowledge, additional exploration is needed.

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
