[Supplementary Material]

# Appendix

---

**Algorithm 1** Q-ThS

---

At time $t$,

Let $\mathsf{E}(t)$ be an independent Bernoulli sample of mean $\min\{1, 3K\frac{\log^2 t}{t}\}$.

**if** $\mathsf{E}(t) = 1$ **then**

   *Explore:*

   Schedule a server uniformly at random.

**else**

   *Exploit:*

   For each $k \in [K]$, pick a sample $\hat{\theta}_k(t)$ of distribution,

$$\hat{\theta}_k(t) \sim \text{Beta}\left(\hat{\mu}_k(t)T_k(t-1) + 1, (1 - \hat{\mu}_k(t))T_k(t-1) + 1\right).$$

   Schedule a server

$$\kappa(t) \in \arg\max_{k \in [K]} \hat{\theta}_k(t).$$

**end if**

---

We present our theoretical results in a more general setting where there are $U$ queues and $K$ servers, such that $1 \leq U \leq K$. All the results in the body of the paper become a special case of this setting when $U = 1$. The queues and servers are indexed by $u = 1, \ldots, U$ and $k = 1, \ldots, K$ respectively. Arrivals to queues and service offered by the links are according to product Bernoulli distribution and i.i.d. across time slots. The mean arrival rates are given by the vector $\boldsymbol{\lambda} = (\lambda_u)_{u \in [U]}$ and the mean service rates by the matrix $\boldsymbol{\mu} = [\mu_{uk}]_{u \in [U], k \in [K]}$.

In any time slot, each server can serve at most one queue and each queue can be served by at most one server. The problem is to schedule, in every time slot, a matching in the complete bipartite graph between queues and servers. The scheduling decision at any time $t$ is based on past observations corresponding to the services obtained for the scheduled matchings until time $t - 1$. Statistical parameters corresponding to the service distributions are considered unknown. The relevant notation for this system has been provided in Table 1.

Table 1: General Notation

| Symbol | Description |
|--------|-------------|
| $\lambda_u$ | Expected rate of arrival to queue $u$ |
| $\lambda_{min}$ | Minimum arrival rate across all queues |
| $A_u(t)$ | Arrival at time $t$ to queue $u$ |
| $\mu_{uk}$ | Expected service rate of server $k$ for queue $u$ |
| $R_{uk}(t)$ | Service rate between server $k$ queue $u$ at time $t$ |
| $k_u^*$ | Best server for queue $u$ |
| $\mu_u^*$ | Expected rate of best server for queue $u$ |
| $\mu_{max}$ | Maximum service rate across all links |
| $\mu_{min}$ | Minimum service rate across all links |
| $\Delta$ | Minimum (among all queues) difference between the best and second best servers |
| $\kappa_u(t)$ | server assigned to queue $u$ at time $t$ |
| $S_u(t)$ | Potential service provided by server assigned to queue $u$ at time $t$ |
| $Q_u(t)$ | queue-length of queue $u$ at time $t$ |
| $Q_u^*(t)$ | queue-length of queue $u$ at time $t$ for the optimal strategy |
| $\Psi_u(t)$ | Regret for queue $u$ at time $t$ |

The queueing system evolution can be described as follows. Let $\kappa_u(t)$ denote the server that is assigned to queue $u$ at time $t$. Therefore, the vector $\boldsymbol{\kappa}(t) = (\kappa_u(t)_{u \in [U]})$ gives the matching scheduled at time $t$. Let $R_{uk}(t)$ be the service offered to queue $u$ by server $k$ and $S_u(t)$ denote the service offered to queue $u$ by server $\kappa_u(t)$ at time $t$. If $\mathbf{A}(t)$ is the (binary) arrival vector at time $t$, then the queue-length vector at time $t$ is given by:

$$\mathbf{Q}(t) = (\mathbf{Q}(t-1) + \mathbf{A}(t) - \mathbf{S}(t))^+ .$$

**Regret Against a Unique Optimal Matching**

Our goal in this paper is to focus attention on how queueing behavior impacts regret minimization in bandit algorithms. To emphasize this point, we consider a somewhat simplified switch scheduling system. In particular, we assume for every queue, there is a unique optimal server with the maximum expected service rate for that queue. Further, we assume that the optimal queue-server pairs form a matching in the complete bipartite graph between queues and servers, that we call the *optimal matching*; and that this optimal matching stabilizes every queue.

Formally, make the following definitions:

$$\mu_u^* := \max_{k \in [K]} \mu_{uk}, \ \ u \in [U]; \tag{3}$$

$$k_u^* := \arg \max_{k \in [K]} \mu_{uk}, \ \ u \in [U]; \tag{4}$$

$$\epsilon_u := \mu_u^* - \lambda_u, \ \ u \in [U]; \tag{5}$$

$$\Delta_{uk} := \mu_u^* - \mu_{uk}, \ \ u \in [U], k \in [K]; \tag{6}$$

$$\Delta := \min_{u \in [U], k \notin k_u^*} \Delta_{uk}; \tag{7}$$

$$\mu_{min} := \min_{u \in [U], k \in [K]} \mu_{uk}; \tag{8}$$

$$\mu_{max} := \max_{u \in [U], k \in [K]} \mu_{uk}; \tag{9}$$

$$\lambda_{min} := \min_{u \in [U]} \lambda_u. \tag{10}$$

The following assumptions will be in force throughout the paper.

**Assumption 2** (Optimal Matching). *There is a* unique optimal matching, *i.e.:*

1. *There is a unique optimal server for each queue: $k_u^*$ is a singleton, i.e., $\Delta_{uk} > 0$ for $k \neq k_u^*$, for all $u$,*

2. *The optimal queue-server pairs for a matching: For any $u' \neq u$, $k_u^* \neq k_{u'}^*$.*

**Assumption 3** (Stability). *The optimal matching stabilizes every queue, i.e., the arrival rates lie within the stability region: $\epsilon_u > 0$ for all $u \in [U]$.*

The assumption of a unique optimal matching essentially means that the queues and servers are solving a pure coordination problem; for example, in the crowdsourcing example described in the introduction, this would correspond to the presence of a unique worker best suited to each type of job. Note that the setting described in Section 3 is equivalent to the unique optimal matching case when $U = 1$. We now describe an algorithm for the unique best match setting which is a more general version of Algorithm 1.

The notation specific to Algorithm 2 has been provided in Table 2.

# 8 Proofs

We provide details of the proofs for Theorem 2 in Section 8.1 and for Theorems 16 and 17 in Section 8.2. In each section, we state and prove a few intermediate lemmas that are useful in proving the theorems.

---

**Algorithm 2** Q-ThS(match)

---

At time $t$,

Let $\mathsf{E}(t)$ be an independent Bernoulli sample of mean $\min\{1, 3K\frac{\log^2 t}{t}\}$.

**if** $\mathsf{E}(t) = 1$ **then**

    *Explore:*

    Schedule a matching from $\mathcal{E}$ uniformly at random.

**else**

    *Exploit:*

    For each $k \in [K], u \in [U]$ , pick a sample $\hat{\theta}_{uk}(t)$ of distribution,

$$\hat{\theta}_{uk}(t) \sim \mathrm{Beta}\left(\hat{\mu}_{uk}(t)T_{uk}(t-1) + 1, (1 - \hat{\mu}_{uk}(t))\, T_{uk}(t-1) + 1\right).$$

    Compute for all $u \in [U]$

$$\hat{k}_u(t) := \arg\max_{k \in [K]} \hat{\theta}_{uk}(t)$$

    Schedule a matching $\boldsymbol{\kappa}(t)$ such that

$$\boldsymbol{\kappa}(t) \in \arg\min_{\boldsymbol{\kappa} \in \mathcal{M}} \sum_{u \in [U]} \mathbb{1}\left\{\kappa_u \neq \hat{k}_u(t)\right\},$$

    i.e., $\boldsymbol{\kappa}(t)$ *is the projection of* $\hat{\boldsymbol{k}}(t)$ *onto the space of all matchings* $\mathcal{M}$ *with Hamming distance as metric.*

**end if**

---

Table 2: Notation specific to Algorithm 2

| Symbol | Description |
|:------:|:-----------|
| $\mathsf{E}(t)$ | Indicates if the algorithm schedules a matching through *Explore* |
| $\mathsf{E}_{uk}(t)$ | Indicates if Server $k$ is assigned to Queue $u$ at time $t$ through *Explore* |
| $\mathsf{I}_{uk}(t)$ | Indicates if Server $k$ is assigned to Queue $u$ at time $t$ through *Exploit* |
| $T_{uk}(t)$ | Number of time slots Server $k$ is assigned to Queue $u$ in time $[1, t]$ |
| $\hat{\boldsymbol{\mu}}(t)$ | Empirical mean of service rates at time $t$ from past observations (until $t-1$) |
| $\boldsymbol{\kappa}(t)$ | Matching scheduled in time-slot $t$ |

## 8.1 Regret Upper Bound for Q-ThS(match)

Theorem 2 is a special case ($U = 1$) of Theorem 6 stated below,

**Theorem 6.** *Consider any problem instance* $(\boldsymbol{\lambda}, \boldsymbol{\mu})$ *which has a single best matching. For any* $u \in [U]$, *let* $w(t) = \exp\left(\left(\frac{2\log t}{\Delta}\right)^{2/3}\right)$, $v'_u(t) = \frac{6K}{\epsilon_u}w(t)$, $t \geq \exp\left(6/\Delta^2\right)$ *and* $v_u(t) = \frac{24}{\epsilon_u^2}\log t + \frac{60K}{\epsilon_u}\frac{v'_u(t)\log^2 t}{t}$. *Then, under Q-ThS(match) the regret for queue $u$,* $\Psi_u(t)$, *satisfies*

$$\Psi_u(t) = O\left(\frac{Kv_u(t)\log^2 t}{t}\right)$$

*for all $t$ such that* $\frac{w(t)}{\log t} \geq \frac{2}{\epsilon_u}$, $t \geq \exp\left(6/\Delta^2\right)$ *and* $v_u(t) + v'_u(t) \leq t/2$.

**Corollary 7.** *Let* $w(t) = \exp\left(\left(\frac{2\log t}{\Delta}\right)^{2/3}\right)$ . *Then,*

$$\Psi_u(t) = O\left(K\frac{\log^3 t}{\epsilon_u^2 t}\right)$$

*for all t such that* $\frac{w(t)}{\log t} \geq \frac{2}{\epsilon_u}$, $\frac{t}{w(t)} \geq \max\left\{\frac{24K}{\epsilon_u}, 15K^2 \log t\right\}$, *and* $\frac{t}{\log t} \geq \frac{198}{\epsilon_u^2}$.

As shown in Algorithm 2, $\mathsf{E}(t)$ indicates whether Q-ThS(match) chooses to explore at time $t$. We now obtain a bound on the expected number of time-slots Q-ThS(match) chooses to explore in an arbitrary time interval $(t_1, t_2]$. Since at any time $t$, Q-ThS(match) decides to explore with probability $\min\{1, 3K\frac{\log^2 t}{t}\}$, we have

$$\mathbb{E}\left[\sum_{l=t_1+1}^{t_2} \mathsf{E}(l)\right] \leq 3K \sum_{l=t_1+1}^{t_2} \frac{\log^2 l}{l} \leq 3K \int_{t_1}^{t_2} \frac{\log^2 l}{l} \, \mathrm{d}l = K\left(\log^3 t_2 - \log^3 t_1\right). \qquad (11)$$

The following lemma gives a probabilistic upper bound on the same quantity.

**Lemma 8.** *For any $t$ and $t_1 < t_2$,*

$$\mathbb{P}\left[\sum_{l=t_1+1}^{t_2} \mathsf{E}(l) \geq 5\max\left(\log t, K\left(\log^3 t_2 - \log^3 t_1\right)\right)\right] \leq \frac{1}{t^4}.$$

*Proof.* To prove the result, we will use the following Chernoff bound: for a sum of independent Bernoulli random variables $Y$ with mean $\mathbb{E}Y$ and for any $\delta > 0$,

$$\mathbb{P}\left[[Y \geq (1+\delta)\mathbb{E}Y] \leq \left(\frac{e^\delta}{(1+\delta)^{1+\delta}}\right)^{\mathbb{E}Y}\right].$$

If $\mathbb{E}Y \geq \log t$, the above bound for $\delta = 4$ gives

$$\mathbb{P}\left[Y \geq 5\mathbb{E}Y\right] \leq \frac{1}{t^4}.$$

Note that $\{\mathsf{E}(l)\}_{l=t_1+1}^{t_2}$ are independent Bernoulli random variables and let $X = \sum_{l=t_1}^{t_2} \mathsf{E}(l)$. Now consider the probability $\mathbb{P}\left[X \geq 5\max\left(\log t, \mathbb{E}X\right)\right]$. If $\mathbb{E}X \geq \log t$, then the result is true from the above Chernoff bound. If $\mathbb{E}X < \log t$, then it is possible to construct a random variable $Y$ which is a sum of independent Bernoulli random variables, has mean $\log t$ and stochastically dominates $X$, in which case we can again use the Chernoff bound on $Y$. Therefore,

$$\mathbb{P}\left[X \geq 5\log t\right] \leq \mathbb{P}\left[Y \geq 5\log t\right] \leq \frac{1}{t^4}.$$

Using inequality (11), we have the required result, i.e.,

$$\mathbb{P}\left[\sum_{l=t_1+1}^{t_2} \mathsf{E}(l) \geq 5\max\left(\log t, K\left(\log^3 t_2 - \log^3 t_1\right)\right)\right] \leq \mathbb{P}\left[X \geq 5\max\left(\log t, \mathbb{E}X\right)\right] \leq 1/t^4.$$

$\square$

Let $w(t) = \exp\left(\left(\frac{2\log t}{\Delta}\right)^{2/3}\right)$. The next lemma shows that, with high probability, Q-ThS(match) does not schedule a sub-optimal matching when it exploits in the late stage.

**Lemma 9.** *For $t \geq \exp\left(6/\Delta^2\right)$,*

$$\mathbb{P}\left[\bigcup_{u\in[U]} \sum_{l=w(t)+1}^{t} \sum_{k\neq k_u^*} \mathsf{I}_{uk}(l) > 0\right] = O\left(\frac{UK}{t^3}\right).$$

*Proof.* Let $X_{uk}(l), u = 1, 2, .., U, k = 1, 2, .., K, l = 1, 2, 3..$ be independent random variables denoting the service offered in the $l^{th}$ assignment of the server $k$ to queue $u$. Consider the events,

$$T_{uk}(w(t)) \geq \frac{1}{2}\log^3(w(t)), \ \forall k \in [K], u \in [U] \qquad (12)$$

$$\theta_{uk_u^*}(s) > \mu_u^* - \sqrt{\frac{\log^2(s)}{T_{uk_u^*}(s)}}, \ \forall s, \ \text{s.t. } w(t)+1 \le s \le t, u \in [U] \tag{13}$$

and

$$\theta_{uk}(s) \le \mu_u^* - \sqrt{\frac{\log^2(s)}{T_{uk_u^*}(s)}}, \ \forall s, k \text{ s.t. } w(t)+1 \le s \le t, k \ne k_u^*, u \in [U] \tag{14}$$

It can be seen that, given the above events, Q-ThS(match) schedules the optimal matching in all time-slots in $(w(t), t]$ in which it decides to exploit, i.e., $\sum_{l=w(t)+1}^{t} \sum_{k \ne k_u^*} l_{uk}(l) = 0$ for all $u \in [U]$. We now show that the events above occur with high probability.

Note that, since the matchings in $\mathcal{E}$ cover all the links in the system, $T_{uk}(w(t)) \le \frac{1}{2}\log^3(w(t))$ for some $u, k$ implies that $\sum_{l=1}^{w(t)} \mathbb{1}\{\boldsymbol{\kappa}(t) = \boldsymbol{\kappa}\} \le \frac{1}{2}\log^3(w(t))$ for some $\boldsymbol{\kappa} \in \mathcal{E}$. Since $\sum_{l=1}^{w(t)} \mathbb{1}\{\boldsymbol{\kappa}(t) = \boldsymbol{\kappa}\}$ is a sum of i.i.d. Bernoulli random variables with mean $\log^3(w(t))$, we use Chernoff bound to prove that event (12) occurs with high probability.

$$\begin{aligned}
\mathbb{P}\left[(12) \text{ is false}\right] &\le \sum_{\boldsymbol{\kappa} \in \mathcal{E}} \mathbb{P}\left[\sum_{l=1}^{w(t)} \mathbb{1}\{\boldsymbol{\kappa}(t) = \boldsymbol{\kappa}\} \le \frac{1}{2}\log^3(w(t))\right] \\
&\le K \exp\left(-\frac{1}{8}\log^3(w(t))\right) \\
&= K \exp\left(-\frac{1}{8}\left(\frac{2\log t}{\Delta}\right)^2\right) = o\left(\frac{K}{t^4}\right).
\end{aligned} \tag{15}$$

In order to prove high probability bounds for the other two events, we define $U_s$ to be a sequence of i.i.d uniform random variables taking values in $[0, 1]$ for $s = w(t)+1, ..., t$. Let us also define $\Sigma_{u,k,l} = \sum_{r=1}^{l} X_{uk}(r)$. In what follows let $F_{a,b}^{\text{Beta}}$ denote the c.d.f of the $\text{Beta}(a, b)$ distribution while $F_{n,p}^{\text{B}}$ denotes the c.d.f. of a $\text{Binomial}(n, p)$ distribution. Let $S_{uk}(t) = \hat{mu}_{uk}(t)T_{uk}(t)$ for all $u \in [U], k \in [K]$.

$$\begin{aligned}
\mathbb{P}\left[(13) \text{ is false}\right] &\le \sum_{u \in [U]} \sum_{s=w(t)+1}^{t} \mathbb{P}\left[\theta_{uk_u^*}(s) \le \mu_u^* - \sqrt{\frac{\log^2(s)}{T_{uk_u^*}(s)}}\right] \\
&= \sum_{u \in [U]} \sum_{s=w(t)+1}^{t} \mathbb{P}\left[U_s \le F_{S_{uk_u^*}(s)+1, T_{uk_u^*}(s)-S_{uk_u^*}(s)+1}^{\text{Beta}}\left(\mu_u^* - \sqrt{\frac{\log^2(s)}{T_{uk_u^*}(s)}}\right)\right] \\
&\overset{(i)}{\le} \sum_{u \in [U]} \sum_{s=w(t)+1}^{t} \mathbb{P}\left[\exists l \in \left\{\frac{1}{2}\log^3(s), ..., s\right\} : F_{l+1, \mu_u^* - \sqrt{\frac{\log^2(s)}{l}}}^{B}\left(\Sigma_{u,k_u^*,l}\right) \le U_s \Big| (12) \text{ is true}\right] \\
&\quad + o\left(\frac{UK}{t^3}\right) \\
&\le \sum_{u \in [U]} \sum_{s=w(t)+1}^{t} \sum_{l=\frac{1}{2}\log^3(s)}^{s} \mathbb{P}\left[\Sigma_{u,k_u^*,l} \le (F^B)^{-1}_{l+1, \mu_u^* - \sqrt{\frac{\log^2(s)}{l}}}(U_s)\right] + o\left(\frac{UK}{t^3}\right)
\end{aligned}$$

In $(i)$ we use the well-known Beta-Binomial trick [] and the fact that given (12) is true, $uk_u^*$ has been scheduled enough number of times. Now the term $(F^B)^{-1}_{l+1, \mu_u^* - \sqrt{\frac{\log^2(s)}{l}}}(U_s)$ can be thought of as

the sum of $l + 1$ i.i.d Bernoulli random variables with mean $\mu_u^* - \sqrt{\frac{\log^2(s)}{l}}$. Let $Z_r$ be a sequence of

i.i.d random variable with mean $\sqrt{\frac{\log^2(s)}{l}}$. Therefore we have,

$$\mathbb{P}\left[\Sigma_{u,k,l} \leq (F^B)^{-1}_{l+1,\mu_u^* - \sqrt{\frac{\log^2(s)}{l}}}(U_s)\right] \leq \mathbb{P}\left[\sum_{r=1}^{l} Z_r \leq 1\right]$$

$$\overset{(ii)}{\leq} e^{-\frac{\log^2(s)}{3}} \tag{16}$$

Here, $(ii)$ is due to Chernoff-Hoeffding's inequality. Therefore we have,

$$\mathbb{P}\left[(13) \text{ is false}\right] \leq U \sum_{s=w(t)+1}^{t} \sum_{l=\frac{1}{2}\log^3(s)}^{s} \exp\left(-\frac{\log^2(s)}{3}\right) + o\left(\frac{UK}{t^3}\right)$$

$$\leq U \exp\left(-\frac{1}{3}\log^2(w(t)) + 2\log t\right) + o\left(\frac{UK}{t^3}\right)$$

$$= U \exp\left(-\frac{1}{3}\left(\frac{2\log t}{\Delta}\right)^{4/3} + 2\log t\right) + o\left(\frac{UK}{t^3}\right) = o\left(\frac{UK}{t^3}\right).$$

$$\mathbb{P}\left[(14) \text{ is false}\right] \leq \sum_{u\in[U], k\neq k_u^*} \sum_{s=w(t)+1}^{t} \mathbb{P}\left[\theta_{uk}(s) > \mu_u^* - \sqrt{\frac{\log^2(s)}{T_{uk_u^*}(s)}}\right]$$

$$\leq \sum_{u\in[U], k\neq k_u^*} \sum_{s=w(t)+1}^{t} \mathbb{P}\left[\theta_{uk}(s) > \mu_u^* - \sqrt{\frac{\log^2(s)}{T_{uk_u^*}(s)}} \middle| (12) \text{ is true}\right] + o\left(\frac{UK}{t^3}\right)$$

$$\overset{(iii)}{\leq} \sum_{u\in[U], k\neq k_u^*} \sum_{s=w(t)+1}^{t} \mathbb{P}\left[\theta_{uk}(s) > \mu_u^* - \sqrt{\frac{2}{\log(s)}} \middle| (12) \text{ is true}\right] + o\left(\frac{UK}{t^3}\right)$$

$$\overset{(iv)}{\leq} \sum_{u\in[U], k\neq k_u^*} \sum_{s=w(t)+1}^{t} \mathbb{P}\left[\theta_{uk}(s) > \mu_{uk} + \frac{\Delta}{2} \middle| (12) \text{ is true}\right] + o\left(\frac{UK}{t^3}\right)$$

$$\overset{(v)}{\leq} \sum_{u\in[U], k\neq k_u^*} \sum_{s=w(t)+1}^{t} \mathbb{P}\left[\exists l \in \left\{\frac{1}{2}\log^3(s), ..., s\right\} : \Sigma_{u,k,l} \geq (F^B)^{-1}_{l+1,\mu_{uk}+\frac{\Delta}{2}}(U_s)\right] + o\left(\frac{UK}{t^3}\right)$$

$$\overset{(vi)}{\leq} o\left(\frac{UK}{t^3}\right)$$

We observe that given (12) is true, we have scheduled $uk_u^*$ enough number of times in order to get $(iii)$. In $(iv)$ we use that fact that $t \geq \exp\left(6/\Delta^2\right)$. $(v)$ is due to the Beta-Binomial trick while $(vi)$ is a result of applying the Chernoff-Hoeffding bound to the first term in $(v)$ in a manner similar to that of (16). $\qquad\square$

For any time $t$, let

$$B_u(t) := \min\{s \geq 0 : Q_u(t-s) = 0\}$$

denote the time elapsed since the beginning of the current regenerative cycle for queue $u$. Alternately, at any time $t$, $t - B_u(t)$ is the last time instant at which queue $u$ was zero.

The following lemma gives an upper bound on the sample-path queue-regret in terms of the number of sub-optimal schedules in the current regenerative cycle.

**Lemma 10.** *For any $t \geq 1$,*

$$Q_u(t) - Q_u^*(t) \leq \sum_{l=t-B_u(t)+1}^{t} \left(\mathsf{E}(l) + \sum_{k\neq k_u^*} \mathsf{I}_{uk}(l)\right).$$

*Proof.* If $B_u(t) = 0$, i.e., if $Q_u(t) = 0$, then the result is trivially true.

Consider the case where $B_u(t) > 0$. Since $Q_u(l) > 0$ for all $t - B_u(t) + 1 \leq l \leq t$, we have

$$Q_u(l) = Q_u(l-1) + A_u(l) - S_u(l) \quad \forall t - B_u(t) + 1 \leq l \leq t.$$

This implies that

$$Q_u(t) = \sum_{l=t-B_u(t)+1}^{t} A_u(l) - S_u(l).$$

Moreover,

$$Q_u^*(t) = \max_{1 \leq s \leq t} \left( Q_u^*(0) + \sum_{l=s}^{t} A_u(l) - S_u^*(l) \right)^+ \geq \sum_{l=t-B_u(t)+1}^{t} A_u(l) - S_u^*(l).$$

Combining the above two expressions, we have

$$Q_u(t) - Q_u^*(t) \leq \sum_{l=t-B_u(t)+1}^{t} S_u^*(l) - S_u(l)$$

$$= \sum_{l=t-B_u(t)+1}^{t} \sum_{k \in [K]} \left( R_{uk_u^*}(l) - R_{uk}(l) \right) \left( \mathsf{E}_{uk}(l) + \mathsf{I}_{uk}(l) \right)$$

$$\leq \sum_{l=t-B_u(t)+1}^{t} \sum_{k \neq k_u^*} \left( \mathsf{E}_{uk}(l) + \mathsf{I}_{uk}(l) \right)$$

$$\leq \sum_{l=t-B_u(t)+1}^{t} \left( \mathsf{E}(l) + \sum_{k \neq k_u^*} \mathsf{I}_{uk}(l) \right),$$

where the second inequality follows from the assumption that the service provided by each of the links is bounded by 1, and the last inequality from the fact that $\sum_{k \in [K]} \mathsf{E}_{uk}(l) = \mathsf{E}(l) \ \forall l, \forall u \in [U]$. □

In the next lemma, we derive a coarse high probability upper bound on the queue-length. This bound on the queue-length is used later to obtain a first cut bound on the length of the regenerative cycle in Lemma 12.

**Lemma 11.** *For any $l \in [1, t]$,*

$$\mathbb{P}\left[ Q_u(l) > 2Kw(t) \right] = O\left( \frac{UK}{t^3} \right)$$

$\forall t$ *s.t.* $\frac{w(t)}{\log t} \geq \frac{2}{\epsilon_u}$ *and* $t \geq \exp\left( 6/\Delta^2 \right)$.

*Proof.* From Lemma 10,

$$Q_u(t) - Q_u^*(t) \leq \sum_{l=t-B_u(t)+1}^{t} \left( \mathsf{E}(l) + \sum_{k \neq k_u^*} \mathsf{I}_{uk}(l) \right) \leq \sum_{l=1}^{t} \left( \mathsf{E}(l) + \sum_{k \neq k_u^*} \mathsf{I}_{uk}(l) \right).$$

Since $Q_u^*(t)$ is distributed according to $\pi_{(\lambda_u, \mu_u^*)}$,

$$\mathbb{P}\left[ Q_u^*(t) > w(t) \right] = \frac{\lambda_u}{\mu_u^*} \left( \frac{\lambda_u (1 - \mu_u^*)}{(1 - \lambda_u) \mu_u^*} \right)^{w(t)} \leq \exp\left( w(t) \log\left( \frac{\lambda_u (1 - \mu_u^*)}{(1 - \lambda_u) \mu_u^*} \right) \right) \leq \frac{1}{t^3}$$

if $\frac{w(t)}{\log t} \geq \frac{2}{\epsilon_u}$. The last inequality follows from the following bound –

$$\log\left( \frac{(1 - \lambda_u) \mu_u^*}{\lambda_u (1 - \mu_u^*)} \right) = \log\left( 1 + \frac{\epsilon_u}{\lambda_u (1 - \mu_u^*)} \right)$$

$$\geq \log\left( 1 + 4\epsilon_u \right) \quad \text{since } \left( \lambda_u (1 - \mu_u^*) < 1/4 \right)$$

$$\geq \frac{3}{2}\epsilon_u.$$

Moreover, from Lemma 8, we have

$$\mathbb{P}\left[\sum_{l=1}^{t}\mathsf{E}(l) > Kw(t)\right] = o\left(\frac{1}{t^3}\right).$$

Now, note that

$$\sum_{l=1}^{t}\sum_{k\neq k_u^*}\mathsf{I}_{uk}(l) \le (K-1)w(t) + \sum_{l=w(t)+1}^{t}\sum_{k\neq k_u^*}\mathsf{I}_{uk}(l).$$

Therefore,

$$\mathbb{P}\left[\sum_{l=1}^{t}\sum_{k\neq k_u^*}\mathsf{I}_{uk}(l) > (K-1)w(t)\right] \le \mathbb{P}\left[\sum_{l=w(t)+1}^{t}\sum_{k\neq k_u^*}\mathsf{I}_{uk}(l) > 0\right] = O\left(\frac{UK}{t^3}\right)$$

from Lemma 9. Using the inequalities above, we have

$$\mathbb{P}\left[Q_u(t) > 2Kw(t)\right] \le \mathbb{P}\left[Q_u^*(t) > w(t)\right] + \mathbb{P}\left[\sum_{l=1}^{t}\mathsf{E}(l) > Kw(t)\right]$$

$$+ \mathbb{P}\left[\sum_{l=1}^{t}\sum_{k\neq k_u^*}\mathsf{I}_{uk}(l) > (K-1)w(t)\right]$$

$$\le \frac{1}{t^3} + O\left(\frac{UK}{t^3}\right)$$

$$= O\left(\frac{UK}{t^3}\right).$$

$\square$

**Lemma 12.** *Let $v_u'(t) = \frac{6K}{\epsilon_u}w(t)$ and let $v_u$ be an arbitrary function. Then,*

$$\mathbb{P}\left[B_u\left(t - v_u(t)\right) > v_u'(t)\right] = O\left(\frac{UK}{t^3}\right)$$

$\forall t$ *s.t.* $\frac{w(t)}{\log t} \ge \frac{2}{\epsilon_u}, t \ge \exp\left(6/\Delta^2\right)$ *and* $v_u(t) + v_u'(t) \le t/2$.

*Proof.* Let $r(t) := t - v_u(t)$. Consider the events

$$Q_u(r(t) - v_u'(t)) \le 2Kw(t), \tag{17}$$

$$\sum_{l=r(t)-v_u'(t)+1}^{r(t)} A_u(l) - R_{uk_u^*}(l) \le -\frac{\epsilon_u}{2}v_u'(t), \tag{18}$$

$$\sum_{l=r(t)-v_u'(t)+1}^{r(t)} \mathsf{E}(l) + \sum_{k\neq k_u^*}\mathsf{I}_{uk}(l) \le Kw(t). \tag{19}$$

By the definition of $v_u'(t)$,

$$2Kw(t) - \frac{\epsilon_u}{2}v_u'(t) \le -Kw(t).$$

Given Events (17)-(19), the above inequality implies that

$$Q_u(r(t) - v_u'(t)) + \sum_{l=r(t)-v_u'(t)+1}^{r(t)} A_u(l) \le \sum_{l=r(t)-v_u'(t)+1}^{r(t)} R_{uk_u^*}(l) - \left(\mathsf{E}(l) + \sum_{k\neq k_u^*}\mathsf{I}_{uk}(l)\right)$$

$$\le \sum_{l=r(t)-v_u'(t)+1}^{r(t)} S_u(l),$$

which further implies that $Q_u(l) = 0$ for some $l \in [r(t) - v'_u(t) + 1, r(t)]$. This gives us that $B_u(r(t)) \leq v'_u(t)$.

We now show that each of the events (17)-(19) occur with high probability. Consider the event (18) and note that $A_u(l) - R_{uk_u^*}(l)$ are i.i.d. random variables with mean $-\epsilon_u$ and bounded between $-1$ and $1$. Using Chernoff bound for sum of bounded i.i.d. random variables, we have

$$\mathbb{P}\left[\sum_{l=r(t)-v'_u(t)+1}^{r(t)} A_u(l) - R_{uk_u^*}(l) > -\frac{\epsilon_u}{2} v'_u(t)\right] \leq \exp\left(-\frac{\epsilon_u^2}{8} v'_u(t)\right) \leq \frac{1}{t^3}$$

since $v'_u(t) \geq \frac{6K}{\epsilon_u} w(t) \geq \frac{24}{\epsilon_u^2} \log t$.

By Lemmas 11, 9 and 8, the probability that any of the events (17), (19) does not occur is $O\left(\frac{UK}{t^3}\right)$ $\forall t$ s.t. $\frac{w(t)}{\log t} \geq \frac{2}{\epsilon_u}$ and $v_u(t) + v'_u(t) \leq t/2$, and therefore we have the required result. $\quad\square$

Using the preceding upper bound on the regenerative cycle-length, we derive tighter bounds on the queue-length and the regenerative cycle-length in Lemmas 14 and 15 respectively. The following lemma is a useful intermediate result.

**Lemma 13.** *For any $u \in [U]$ and $t_2$ s.t. $1 \leq t_2 \leq t$,*

$$\mathbb{P}\left[\max_{1 \leq s \leq t_2}\left\{\sum_{l=t_2-s+1}^{t_2} A_u(l) - R_{uk_u^*}(l)\right\} \geq \frac{2\log t}{\epsilon_u}\right] \leq \frac{1}{t^3}.$$

*Proof.* Let $X_s = \sum_{l=t_2-s+1}^{t_2} A_u(l) - R_{uk_u^*}(l)$. Since $X_s$ is the sum of $s$ i.i.d. random variables with mean $\epsilon_u$ and is bounded within $[-1, 1]$, Hoeffding's inequality gives

$$\mathbb{P}\left[X_s \geq \frac{2\log t}{\epsilon_u}\right] = \mathbb{P}\left[X_s - \mathbb{E}X_s \geq \epsilon_u s + \frac{2\log t}{\epsilon_u}\right]$$

$$\leq \exp\left(-\frac{2\left(\epsilon_u s + \frac{2\log t}{\epsilon_u}\right)^2}{4s}\right)$$

$$\leq \exp\left(-4\log t\right),$$

where the last inequality follows from the fact that $(a + b)^2 > 4ab$ for any $a, b \geq 0$. Using union bound over all $1 \leq s \leq t_2$ gives the required result. $\quad\square$

**Lemma 14.** *Let $v'_u(t) = \frac{6K}{\epsilon_u} w(t)$ and $v_u$ be an arbitrary function. Then,*

$$\mathbb{P}\left[Q_u(t - v_u(t)) > \left(\frac{2}{\epsilon_u} + 5\right)\log t + 30K\frac{v'_u(t)\log^2 t}{t}\right] = O\left(\frac{UK}{t^3}\right)$$

$\forall t$ s.t. $\frac{w(t)}{\log t} \geq \frac{2}{\epsilon_u}, t \geq \exp\left(6/\Delta^2\right)$ *and* $v_u(t) + v'_u(t) \leq t/2$.

*Proof.* Let $r(t) = t - v_u(t)$. Now, consider the events

$$B_u(r(t)) \leq v'_u(t), \tag{20}$$

$$\sum_{l=r(t)-s+1}^{r(t)} A_u(l) - R_{uk_u^*}(l) \leq \frac{2\log t}{\epsilon_u} \quad 1 \leq s \leq v'_u(t), \tag{21}$$

$$\sum_{l=r(t)-v'_u(t)+1}^{r(t)} \mathsf{E}(l) + \sum_{k \neq k_u^*} \mathsf{I}_{uk}(l) \leq 5\log t + 5K\left(\log^3\left(r(t)\right) - \log^3\left(r(t) - v'_u(t)\right)\right). \tag{22}$$

Given the above events, we have

$$Q_u(r(t)) = \sum_{l=r(t)-B_u(r(t))+1}^{r(t)} A_u(l) - S(l)$$

$$\leq \sum_{l=r(t)-B_u(r(t))+1}^{r(t)} A_u(l) - R_{uk_u^*}(l) + \mathsf{E}(l) + \sum_{k\neq k^*} \mathsf{I}_{uk}(l)$$

$$\leq \left(\frac{2}{\epsilon_u} + 5\right) \log t + 5K \left(\log^3(r(t)) - \log^3(r(t) - v_u'(t))\right)$$

$$\leq \left(\frac{2}{\epsilon_u} + 5\right) \log t + 15K \frac{v_u'(t)\log^2 t}{(r(t) - v_u'(t))}$$

$$\leq \left(\frac{2}{\epsilon_u} + 5\right) \log t + 30K \frac{v_u'(t)\log^2 t}{t},$$

where the last inequality is true if $v_u(t) + v_u'(t) \leq t/2$. From Lemmas 12, 13, 9 and 8, probability of each the events (20)-(22) is $1 - O\left(\frac{UK}{t^3}\right)$ and therefore, we have the required result. $\square$

**Lemma 15.** *Let* $v_u'(t) = \frac{6K}{\epsilon_u} w(t)$ *and* $v_u(t) = \frac{24\log t}{\epsilon_u^2} + \frac{60K}{\epsilon_u}\frac{v_u'(t)\log^2 t}{t}$. *Then,*

$$\mathbb{P}\left[B_u(t) > v_u(t)\right] = O\left(\frac{UK}{t^3}\right)$$

$\forall t$ *s.t.* $\frac{w(t)}{\log t} \geq \frac{2}{\epsilon_u}, t \geq \exp\left(6/\Delta^2\right)$ *and* $v_u(t) + v_u'(t) \leq t/2$.

*Proof.* Let $r(t) = t - v_u(t)$. As in Lemma 12, consider the events

$$Q_u(r(t)) \leq \left(\frac{2}{\epsilon_u} + 5\right) \log t + 30K \frac{v_u'(t)\log^2 t}{t}, \tag{23}$$

$$\sum_{l=r(t)+1}^{t} A_u(l) - R_{uk_u^*}(l) \leq -\frac{\epsilon_u}{2} v_u(t), \tag{24}$$

$$\sum_{l=r(t)+1}^{t} \mathsf{E}(l) + \sum_{k\neq k_u^*} \mathsf{I}_{uk}(l) \leq 5\log t + 5K \left(\log^3 t - \log^3(r(t))\right). \tag{25}$$

The definition of $v_u(t)$ and events (23)-(25) imply that

$$Q_u(r(t)) + \sum_{l=r(t)+1}^{t} A_u(l) \leq \sum_{l=r(t)+1}^{t} R_{uk_u^*}(l) - \sum_{l=r(t)+1}^{t} \mathsf{E}(l) + \sum_{k\neq k_u^*} \mathsf{I}_{uk}(l)$$

$$\leq \sum_{l=r(t)+1}^{t} S_u(l),$$

which further implies that $Q(l) = 0$ for some $l \in [r(t)+1, t]$ and therefore $B_u(t) \leq v_u(t)$. We can again show that each of the events (23)-(25) occurs with high probability. Particularly, by Lemmas 8, 9 and 14, the probability that any one of the events (23), (25) does not occur is $O\left(\frac{UK}{t^3}\right)$ $\forall t$ s.t. $\frac{w(t)}{\log t} \geq \frac{2}{\epsilon_u}$ and $v_u(t) + v_u'(t) \leq t/2$. We can bound the probability of event (24) in the same way as event (21) in Lemma 12 to show that it occurs with probability at least $\frac{1}{t^3}$. Combining all these gives us the required high probability result. $\square$

*Proof of Theorem 6.* The proof is based on two main ideas: one is that the regenerative cycle length is not very large, and the other is that the algorithm has correctly identified the optimal matching

in late stages. We combine Lemmas 9 and 15 to bound the regret at any time $t$ s.t. $\frac{w(t)}{\log t} \geq \frac{2}{\epsilon_u}$ and $v_u(t) + v'_u(t) \leq t/2$:

$$\Psi_u(t) = \mathbb{E}\left[Q_u(t) - Q_u^*(t)\right]$$

$$\leq \mathbb{E}\left[Q_u(t) - Q_u^*(t)\bigg| B_u(t) \leq v_u(t)\right] \mathbb{P}\left[B_u(t) \leq v_u(t)\right]$$

$$+ \mathbb{E}\left[Q_u(t) - Q_u^*(t)\bigg| B_u(t) > v_u(t)\right] \mathbb{P}\left[B_u(t) > v_u(t)\right]$$

$$\leq \mathbb{E}\left[\sum_{l=t-v_u(t)+1}^{t} \mathsf{E}(l) + \sum_{k \neq k_u^*} \mathsf{I}_{uk}(l)\right] + t\mathbb{P}\left[B_u(t) > v_u(t)\right] \tag{26}$$

$$\leq K\left(\log^3(t) - \log^3(t - v_u(t))\right) + t\mathbb{P}\left[\sum_{l=t-v_u(t)+1}^{t}\sum_{k \neq k_u^*} \mathsf{I}_{uk}(l) > 0\right] + t\mathbb{P}\left[B_u(t) > v_u(t)\right] \tag{27}$$

$$\leq 3K \log^2 t \log\left(1 + \frac{v_u(t)}{t - v_u(t)}\right) + O\left(\frac{UK}{t^2}\right)$$

$$= O\left(K\frac{v_u(t) \log^2 t}{t - v_u(t)}\right) + O\left(\frac{U}{tw(t)}\right)$$

$$= O\left(K\frac{v_u(t) \log^2 t}{t}\right),$$

where (26) follows from Lemma 10, and the last two terms in inequality (27) are bounded using Lemmas 9 and 15. $\qquad\square$

*Proof of Corollary 7.* We first note the following:

(i) $\frac{t}{w(t)} \geq \frac{24K}{\epsilon_u}$ implies that $v'_u(t) \leq \frac{t}{4}$ ,

(ii) $\frac{t}{w(t)} \geq 15K^2 \log t$ implies that $\frac{24}{\epsilon_u^2} \log t \geq \frac{60K}{\epsilon_u}\frac{v'_u(t) \log^2 t}{t}$, and therefore $v_u(t) \leq \frac{48}{\epsilon_u^2} \log t$

(iii) $\frac{t}{\log t} \geq \frac{198}{\epsilon_u^2}$ implies that $v_u(t) \leq \frac{t}{4}$.

These inequalities when applied to Theorem 6 give the required result. $\qquad\square$

## 8.2 Lower Bounds for $\alpha$-Consistent Policies

As mentioned earlier, we prove asymptotic and early stage lower bounds for a class of policies called the $\alpha$-consistent class (Definition 1). As before we will be proving our results for a more general case where there are $U$ queues and $K$ servers. Theorems 1 and 4 are special cases of the analogous theorems stated below, under the unique optimal matching assumption.

**Theorem 16.** *For any problem instance* $(\boldsymbol{\lambda}, \boldsymbol{\mu})$ *with a unique optimal matching, and any $\alpha$-consistent policy, the regret* $\boldsymbol{\Psi}(t)$ *satisfies*

*(a)*

$$\frac{1}{U}\sum_{u \in [U]} \Psi_u(t) \geq \left(\frac{\lambda_{min}}{8} D(\boldsymbol{\mu})(1 - \alpha)(K - 1)\right)\frac{1}{t},$$

*(b) and for any $u \in [U]$,*

$$\Psi_u(t) \geq \left(\frac{\lambda_{min}}{8} D(\boldsymbol{\mu})(1 - \alpha)\max\left\{U - 1, 2(K - U)\right\}\right)\frac{1}{t}$$

*for infinitely many t, where*

$$D(\boldsymbol{\mu}) = \frac{\Delta}{\mathrm{KL}\left(\mu_{min}, \frac{\mu_{max}+1}{2}\right)}. \tag{28}$$

**Theorem 17.** *Given any problem instance $(\boldsymbol{\lambda}, \boldsymbol{\mu})$, and for any $\alpha$-consistent policy and $\gamma > \frac{1}{1-\alpha}$, the regret $\boldsymbol{\Psi}(t)$ satisfies*

*(a)*

$$\frac{1}{U} \sum_{u \in [U]} \Psi_u(t) \geq \frac{D(\boldsymbol{\mu})}{4}(K-1)\frac{\log t}{\log \log t},$$

*for $t \in \left[\max\{C_4 K^\gamma, \tau\}, (K-1)\frac{D(\boldsymbol{\mu})}{4\bar{\epsilon}}\right]$, and*

*(b) for any $u \in [U]$,*

$$\Psi_u(t) \geq \frac{D(\boldsymbol{\mu})}{4} \max\{U-1, 2(K-U)\} \frac{\log t}{\log \log t}$$

*for $t \in \left[\max\{C_4 K^\gamma, \tau\}, (K-1)\frac{D(\boldsymbol{\mu})}{2\epsilon_u}\right]$,*

*where $D(\boldsymbol{\mu})$ is given by equation 28, $\bar{\epsilon} = \frac{1}{U}\sum_{u \in [U]} \epsilon_u$, and $\tau$ and $C_4$ are constants that depend on $\alpha$, $\gamma$ and the policy.*

In order to prove Theorems 16 and 17, we use techniques from existing work in the MAB literature along with some new lower bounding ideas specific to queueing systems. Specifically, we use lower bounds for $\alpha$-consistent policies on the expected number of times a sub-optimal server is scheduled. This lower bound, shown (in Lemma 19) specifically for the problem of scheduling a unique optimal matching, is similar in style to the traditional bandit lower bound by Lai et al. [7] but holds in the non-asymptotic setting. Also, as opposed the traditional change of measure proof technique used in [7], the proof (similar to the more recent ones [21, 22, 19]) uses results from hypothesis testing (Lemma 18).

**Lemma 18** ([23]). *Consider two probability measures $P$ and $Q$, both absolutely continuous with respect to a given measure. Then for any event $\mathcal{A}$ we have:*

$$P(\mathcal{A}) + Q(\mathcal{A}^c) \geq \frac{1}{2} \exp\{-\min(\mathrm{KL}(P||Q), \mathrm{KL}(Q||P))\}.$$

*Proof.* Let $p = P(\mathcal{A})$ and $q = Q(\mathcal{A}^c)$. From standard properties of KL divergence we have that,

$$\mathrm{KL}(P||Q) \geq \mathrm{KL}(p, q)$$

Therefore, it is sufficient to prove that

$$p + q \geq \frac{1}{2} \exp\left(-p \log \frac{p}{1-q} - (1-p) \log \frac{1-p}{q}\right) = \frac{1}{2}\left(\frac{1-q}{p}\right)^p \left(\frac{q}{1-p}\right)^{1-p}.$$

Now,

$$\begin{aligned}
\left(\frac{1-q}{p}\right)^p \left(\frac{q}{1-p}\right)^{1-p} &= \left(\sqrt{\frac{1-q}{p}}\right)^{2p} \left(\sqrt{\frac{q}{1-p}}\right)^{2(1-p)} \\
&\leq \left(\frac{1}{2}\left(2p \cdot \sqrt{\frac{1-q}{p}} + 2(1-p) \cdot \sqrt{\frac{q}{1-p}}\right)\right)^2 \\
&= \left(\sqrt{p(1-q)} + \sqrt{q(1-p)}\right)^2 \\
&\leq 2(p(1-q) + q(1-p)) \\
&< 2(p+q)
\end{aligned}$$

as required. $\square$

**Lemma 19.** *For any problem instance* $(\boldsymbol{\lambda}, \boldsymbol{\mu})$ *and any* $\alpha$-*consistent policy, there exist constants* $\tau$ *and* $C$ *s.t. for any* $u \in [U]$, $k \neq k_u^*$ *and* $t > \tau$,

$$\mathbb{E}\left[T_{uk}(t)\right] + \sum_{u' \neq u} \mathbb{1}\left\{k_{u'}^* = k\right\} \mathbb{E}\left[T_{u'k_u^*}(t)\right] \geq \frac{1}{\text{KL}\left(\mu_{min}, \frac{\mu_{max}+1}{2}\right)} \left((1-\alpha)\log t - \log(4KC)\right).$$

*Proof.* Without loss of generality, let the optimal servers for the $U$ queues be denoted by the first $U$ indices. In other words, a server $k > U$ is not an optimal server for any queue, i.e., for any $u' \in [U]$, $K \geq k > U$, $\mathbb{1}\left\{k_{u'}^* = k\right\} = 0$. Also, let $\beta = \frac{\mu_{max}+1}{2}$.

We will first consider the case $k \leq U$. For a fixed user $u$ and server $k \leq U$, let $u'$ be the queue that has $k$ as the best server, i.e., $k_{u'}^* = k$. Now consider the two problem instances $(\boldsymbol{\lambda}, \boldsymbol{\mu})$ and $(\boldsymbol{\lambda}, \hat{\boldsymbol{\mu}})$, where $\hat{\boldsymbol{\mu}}$ is the same as $\boldsymbol{\mu}$ except for the two entries corresponding to indices $(u, k)$, $(u', k_u^*)$ replaced by $\beta$. Therefore, for the problem instance $(\boldsymbol{\lambda}, \hat{\boldsymbol{\mu}})$, the best servers are swapped for queues $u$ and $u'$ and remain the same for all the other queues. Let $\mathbb{P}_{\boldsymbol{\mu}}^t$ and $\mathbb{P}_{\hat{\boldsymbol{\mu}}}^t$ be the distributions corresponding to the arrivals, chosen servers and rates obtained in the first $t$ plays for the two instances under a fixed $\alpha$-consistent policy. Recall that $T_{uk}(t) = \sum_{s=1}^t \mathbb{1}\{\kappa_u(s) = k\} \; \forall u \in [U], k \in [K]$. Define the event $\mathcal{A} = \{T_{uk}(t) > t/2\}$. By the definition of $\alpha$-consistency there exists a fixed integer $\tau$ and a fixed constant $C$ such that for all $t > \tau$ we have,

$$\mathbb{E}_{\boldsymbol{\mu}}^t \left[\sum_{s=1}^t \mathbb{1}\{\kappa_u(s) = k\}\right] \leq Ct^\alpha$$

$$\mathbb{E}_{\hat{\boldsymbol{\mu}}}^t \left[\sum_{s=1}^t \mathbb{1}\{\kappa_u(s) = k'\}\right] \leq Ct^\alpha, \forall k' \neq k.$$

A simple application of Markov's inequality yields

$$\mathbb{P}_{\boldsymbol{\mu}}^t(\mathcal{A}) \leq \frac{2C}{t^{1-\alpha}}$$

$$\mathbb{P}_{\hat{\boldsymbol{\mu}}}^t(\mathcal{A}^c) \leq \frac{2C(K-1)}{t^{1-\alpha}}.$$

We can now use Lemma 18 to conclude that

$$\text{KL}(\mathbb{P}_{\boldsymbol{\mu}}^t || \mathbb{P}_{\hat{\boldsymbol{\mu}}}^t) \geq (1-\alpha)\log t - \log(4KC). \tag{29}$$

It is, therefore, sufficient to show that

$$\text{KL}\left(\mathbb{P}_{\boldsymbol{\mu}}^t || \mathbb{P}_{\hat{\boldsymbol{\mu}}}^t\right) = \text{KL}\left(\mu_{uk}, \beta\right) \mathbb{E}_{\boldsymbol{\mu}}^t[T_{uk}(t)] + \text{KL}\left(\mu_{u'k_u^*}, \beta\right) \mathbb{E}_{\boldsymbol{\mu}}^t[T_{u'k_u^*}(t)].$$

For the sake of brevity we write the scheduling sequence in the first $t$ time-slots $\{\boldsymbol{\kappa}(1), \boldsymbol{\kappa}(2), ..., \boldsymbol{\kappa}(t)\}$ as $\boldsymbol{\kappa}^{(t)}$, and similarly we define $\mathbf{A}^{(t)}$ as the number of arrivals to the queue and $\mathbf{S}^{(t)}$ as the service offered by the scheduled servers in the first $t$ time-slots. Let $\mathbf{Z}^{(t)} = (\boldsymbol{\kappa}^{(t)}, \mathbf{A}^{(t)}, \mathbf{S}^{(t)})$. The KL-divergence term can now be written as

$$\text{KL}(\mathbb{P}_{\boldsymbol{\mu}}^t || \mathbb{P}_{\hat{\boldsymbol{\mu}}}^t) = \text{KL}(\mathbb{P}_{\boldsymbol{\mu}}^t(\mathbf{Z}^{(t)}) || \mathbb{P}_{\hat{\boldsymbol{\mu}}}^t(\mathbf{Z}^{(t)})).$$

We can apply the chain rule of divergence to conclude that

$$\text{KL}(\mathbb{P}_{\boldsymbol{\mu}}^t(\mathbf{Z}^{(t)}) || \mathbb{P}_{\hat{\boldsymbol{\mu}}}^t(\mathbf{Z}^{(t)})) = \text{KL}(\mathbb{P}_{\boldsymbol{\mu}}^t(\mathbf{Z}^{(t-1)}) || \mathbb{P}_{\hat{\boldsymbol{\mu}}}^t(\mathbf{Z}^{(t-1)}))$$
$$+ \text{KL}(\mathbb{P}_{\boldsymbol{\mu}}^t(\boldsymbol{\kappa}(t) \mid \mathbf{Z}^{(t-1)}) || \mathbb{P}_{\hat{\boldsymbol{\mu}}}^t(\boldsymbol{\kappa}(t) \mid \mathbf{Z}^{(t-1)}))$$
$$+ \mathbb{E}_{\boldsymbol{\mu}}^t \left[\mathbb{1}\{\kappa_u(t) = k\}\text{KL}\left(\mu_{uk}, \beta\right) + \mathbb{1}\{\kappa_{u'}(t) = k_u^*\}\text{KL}\left(\mu_{u'k_u^*}, \beta\right)\right].$$

We can apply this iteratively to obtain

$$\text{KL}(\mathbb{P}_{\boldsymbol{\mu}}^t || \mathbb{P}_{\hat{\boldsymbol{\mu}}}^t) = \sum_{s=1}^t \mathbb{E}_{\boldsymbol{\mu}}^t \left[\mathbb{1}\{\kappa_u(s) = k\}\text{KL}\left(\mu_{uk}, \beta\right)\right]$$

$$+ \sum_{s=1}^t \mathbb{E}_{\boldsymbol{\mu}}^t \left[\mathbb{1}\{\kappa_{u'}(s) = k_u^*\}\text{KL}\left(\mu_{u'k_u^*}, \beta\right)\right]$$

$$+ \sum_{l=1}^t \text{KL}(\mathbb{P}_{\boldsymbol{\mu}}^t(\boldsymbol{\kappa}(l) \mid \mathbf{Z}^{(l-1)}) || \mathbb{P}_{\hat{\boldsymbol{\mu}}}^t(\boldsymbol{\kappa}(l) \mid \mathbf{Z}^{(l-1)})) \tag{30}$$

Note that the second summation in (30) is zero, as over a sample path the policy pulls the same servers irrespective of the parameters. Therefore, we obtain

$$\text{KL}(\mathbb{P}^t_{\boldsymbol{\mu}}||\mathbb{P}^t_{\hat{\boldsymbol{\mu}}}) = \text{KL}\left(\mu_{uk}, \beta\right)\mathbb{E}^t_{\boldsymbol{\mu}}[T_{uk}(t)] + \text{KL}\left(\mu_{u'k^*_u}, \beta\right)\mathbb{E}^t_{\boldsymbol{\mu}}[T_{u'k^*_u}(t)],$$

which can be substituted in (29) to obtain the required result for $K \leq U$.

Now, consider the case $k > U$, where $\sum_{u \in U} \mathbb{1}\left\{k^*_u = k\right\} = 0$. We again compare the two problem instances $(\boldsymbol{\lambda}, \boldsymbol{\mu})$ and $(\boldsymbol{\lambda}, \hat{\boldsymbol{\mu}})$, where $\hat{\boldsymbol{\mu}}$ is the same as $\boldsymbol{\mu}$ except for the entry corresponding to index $(u, k)$ replaced by $\beta$. Therefore, for the problem instance $(\boldsymbol{\lambda}, \hat{\boldsymbol{\mu}})$, the best server for user $u$ is server $k$ while the best servers for all other queues remain the same. We can again use the same technique as before to obtain

$$\text{KL}(\mathbb{P}^t_{\boldsymbol{\mu}}||\mathbb{P}^t_{\hat{\boldsymbol{\mu}}}) = \text{KL}\left(\mu_{uk}, \beta\right)\mathbb{E}^t_{\boldsymbol{\mu}}[T_{uk}(t)],$$

which, along with (29), gives the required result for $K > U$. $\qquad\square$

As a corollary of the above result, we now derive lower bound on the total expected number of sub-optimal schedules summed across all queues. In addition, we also show, for each individual queue, a lower bound for those servers which are sub-optimal for all the queues. As in the proof of Lemma 19, we assume without loss of generality that the first $U$ indices denote the optimal servers for the $U$ queues.

**Corollary 20.** *For any problem instance $(\boldsymbol{\lambda}, \boldsymbol{\mu})$ and any $\alpha$-consistent policy, there exist constants $\tau$ and $C$ s.t. for any $t > \tau$,*

*(a)*

$$2\Delta \sum_{u \in [U]} \sum_{k \neq k^*_u} \mathbb{E}\left[T_{uk}(t)\right] \geq U(K-1)D(\boldsymbol{\mu})\left((1-\alpha)\log t - \log(4KC)\right),$$

*(b) for any $u \in [U]$,*

$$2\Delta \sum_{k \neq k^*_u} \mathbb{E}\left[T_{uk}(t)\right] \geq (U-1)D(\boldsymbol{\mu})\left((1-\alpha)\log t - \log(4KC)\right),$$

*(c) and for any $u \in [U]$,*

$$\Delta \sum_{k > U} \mathbb{E}\left[T_{uk}(t)\right] \geq (K-U)D(\boldsymbol{\mu})\left((1-\alpha)\log t - \log(4KC)\right),$$

*where $D(\boldsymbol{\mu})$ is given by (28).*

*Proof.* To prove part (a), we observe that a unique optimal server for each queue in the system implies that

$$\sum_{u \in [U]} \sum_{k \neq k^*_u} \mathbb{E}\left[T_{uk}(t)\right] \geq \sum_{u \in [U]} \sum_{u' \neq u} \mathbb{E}\left[T_{uk^*_{u'}}(t)\right]$$

$$= \sum_{u \in [U]} \sum_{k \neq k^*_u} \sum_{u' \neq u} \mathbb{1}\left\{k^*_{u'} = k\right\}\mathbb{E}\left[T_{u'k^*_u}(t)\right].$$

Now, from Lemma 19, there exist constants $C$ and $\tau$ such that for $t > \tau$,

$$2\sum_{u \in [U]} \sum_{k \neq k^*_u} \mathbb{E}\left[T_{uk}(t)\right] \geq \sum_{u \in [U]} \sum_{k \neq k^*_u}\left(\mathbb{E}\left[T_{uk}(t)\right] + \sum_{u' \neq u} \mathbb{1}\left\{k^*_{u'} = k\right\}\mathbb{E}\left[T_{u'k^*_u}(t)\right]\right)$$

$$\geq \frac{U(K-1)}{\text{KL}\left(\mu_{min}, \frac{\mu_{max}+1}{2}\right)}\left((1-\alpha)\log t - \log(4KC)\right).$$

Using the definition of $D(\boldsymbol{\mu})$ in the above inequality gives part (a) of the corollary.

To prove part (b), we can assume without loss of generality that a perfect matching is scheduled in every time-slot. Using this, and the fact that any server is assigned to at most one queue in every time-slot, for any $u \in [U]$, we have

$$T_{uk_u^*}(t) + \sum_{k \neq k_u^*} T_{uk}(t) = t \geq T_{uk_u^*}(t) + \sum_{u' \neq u} T_{u'k_u^*}(t),$$

which gives us

$$\sum_{k \neq k_u^*} T_{uk}(t) \geq \max \left\{ \sum_{u' \neq u} T_{uk_{u'}^*}(t), \sum_{u' \neq u} T_{u'k_u^*}(t) \right\}. \tag{31}$$

From Lemma 19 we have, for any $u' \neq u$ and for $t > \tau$,

$$\mathbb{E}\left[T_{uk_{u'}^*}(t)\right] + \mathbb{E}\left[T_{u'k_u^*}(t)\right] \geq \frac{1}{\mathrm{KL}\left(\mu_{min}, \frac{\mu_{max}+1}{2}\right)} \left((1-\alpha)\log t - \log(4KC)\right),$$

which gives

$$\sum_{u' \neq u} \mathbb{E}\left[T_{uk_{u'}^*}(t)\right] + \mathbb{E}\left[T_{u'k_u^*}(t)\right] \geq \frac{U-1}{\mathrm{KL}\left(\mu_{min}, \frac{\mu_{max}+1}{2}\right)} \left((1-\alpha)\log t - \log(4KC)\right).$$

Combining the above with (31), we have for $t > \tau$

$$\sum_{k \neq k_u^*} \mathbb{E}\left[T_{uk}(t)\right] \geq \max \left\{ \sum_{u' \neq u} \mathbb{E}\left[T_{uk_{u'}^*}(t)\right], \sum_{u' \neq u} \mathbb{E}\left[T_{u'k_u^*}(t)\right] \right\}$$

$$\geq \frac{U-1}{2\mathrm{KL}\left(\mu_{min}, \frac{\mu_{max}+1}{2}\right)} \left((1-\alpha)\log t - \log(4KC)\right).$$

To prove part (c), we use the fact that $\mathbb{1}\left\{k_{u'}^* = k\right\} = 0$ for any $u' \in [U]$, $K \geq k > U$. Therefore, for $t > \tau$, we have

$$\sum_{k > U} \mathbb{E}\left[T_{uk}(t)\right] = \sum_{k > U} \left( \mathbb{E}\left[T_{uk}(t)\right] + \sum_{u' \neq u} \mathbb{1}\left\{k_{u'}^* = k\right\} \mathbb{E}\left[T_{u'k_u^*}(t)\right] \right)$$

$$\geq \frac{K-U}{\mathrm{KL}\left(\mu_{min}, \frac{\mu_{max}+1}{2}\right)} \left((1-\alpha)\log t - \log(4KC)\right),$$

which gives the required result. □

### 8.2.1 Late Stage: Proof of Theorem 16

The following lemma, which gives a lower bound on the queue-regret in terms of probability of sub-optimal schedule in a single time-slot, is the key result used in the proof of Theorem 16. The proof for this lemma is based on the idea that the growth in regret in a single-time slot can be lower bounded in terms of the probability of sub-optimal schedule in that time-slot.

**Lemma 21.** *For any problem instance characterized by $(\boldsymbol{\lambda}, \boldsymbol{\mu})$, and for any scheduling policy, and user $u \in [U]$,*

$$\Psi_u(t) \geq \lambda_u \sum_{k \neq k_u^*} \Delta_{uk} \mathbb{P}\left[\mathbb{1}\{\kappa_u(t) = k\} = 1\right].$$

*Proof.* For the given queueing system, consider an alternate coupled queueing system such that

1. the two systems start with the same initial condition,

2. the arrival process for both the systems is the same, and

3. the service process for the alternate system is independent of the arrival process and i.i.d. across time-slots. For each queue in the alternate system, the service offered by different servers at any time-slot could possibly be dependent on each other but has the same marginal distribution as that in the original system and is independent of the service offered to other queues.

We first show that, under any scheduling policy, the regret for the alternate system has the same distribution as that for the original system. Note that the evolution of the queues is a function of the process $(\mathbf{Z}(l))_{l\geq1} := (\mathbf{A}(l), \boldsymbol{\kappa}(l), \mathbf{S}(l))_{l\geq1}$. To prove that this process has the same distribution in both the systems, we use induction on the size of the finite-dimensional distribution of the process. In other words, we show that the distribution of the vector $(\mathbf{Z}(l))_{l=1}^{t}$ is the same for the two systems for all $t$ by induction on $t$.

Suppose that the hypothesis is true for $t-1$. Now consider the conditional distribution of $\mathbf{Z}(t)$ given $(\mathbf{Z}(l))_{l=1}^{t-1}$. Given $(\mathbf{Z}(l))_{l=1}^{t-1}$, the distribution of $(\mathbf{A}(t), \boldsymbol{\kappa}(t))$ is identical for the two systems for any scheduling policy since the two systems have the same arrival process. Also, given $\left((\mathbf{Z}(l))_{l=1}^{t-1}, \mathbf{A}(t), \boldsymbol{\kappa}(t)\right)$, the distribution of $\mathbf{S}(t)$ depends only on the marginal distribution of the scheduled servers given by $\boldsymbol{\kappa}(t)$ which is again the same for the two systems. Therefore, $(\mathbf{Z}(l))_{l=1}^{t}$ has the same distribution in the two systems. Since the statement is true for $t=1$, it is true for all $t$.

Thus, to lower bound the queue-regret for any queue $u \in [U]$ in the original system, it is sufficient to lower bound the corresponding queue-regret of an alternate queueing system constructed as follows: let $\{U(t)\}_{t\geq1}$ be i.i.d. random variables distributed uniformly in $(0,1)$. For the alternate system, let the service process for queue $u$ and server $k$ be given by $R_{uk}(t) = \mathbb{1}\left\{U(t) \leq \mu_{uk}\right\}$. Since $\mathbb{E}\left[R_{uk}(t)\right] = \mu_{uk}$, the marginals of the service offered by each of the servers is the same as the original system. In addition, the initial condition, the arrival process and the service process for all other queues in the alternate system are identical to those in the original system.

We now lower bound the queue-regret for queue $u$ in the alternate system. Note that, since $\mu_u^* > \mu_{uk}$ $\forall k \neq k_u^*$, we have $R_{uk_u^*}(t) \geq R_{uk}(t)$ $\forall k \neq k_u^*$, $\forall t$. This implies that $Q_u^*(t) \leq Q_u(t)$ $\forall t$. Now, for any given $t$, using the fact that $Q_u^*(t-1) \leq Q_u(t-1)$, it is easy to see that

$$Q_u(t) - Q_u^*(t) \geq \mathbb{1}\left\{A_u(t) = 1\right\}\left(R_{k_u^*}(t) - \sum_{k=1}^{K} \mathbb{1}\{\kappa_u(t) = k\} R_{uk}(t)\right).$$

Therefore,

$$\mathbb{E}\left[Q_u(t) - Q_u^*(t)\right] \geq \mathbb{E}\left[\mathbb{1}\{A_u(t) = 1\}\left(R_{k_u^*}(t) - \sum_{k=1}^{K} \mathbb{1}\{\kappa_u(t) = k\} R_{uk}(t)\right)\right]$$

$$= \lambda_u \sum_{k \neq k_u^*} \mathbb{P}\left[\mathbb{1}\{\kappa_u(t) = k\} = 1\right] \mathbb{P}\left[\mu_{uk} < U(t) \leq \mu_u^*\right]$$

$$= \lambda_u \sum_{k \neq k_u^*} \Delta_{uk} \mathbb{P}\left[\mathbb{1}\{\kappa_u(t) = k\} = 1\right].$$

$\square$

We now use Lemma 21 in conjunction with the lower bound for the expected number of sub-optimal schedules for an $\alpha$-consistent policy (Corollary 20) to prove Theorem 16.

*Proof of Theorem 16.* From Lemma 21 we have,

$$\Psi_u(t) \geq \lambda_u \sum_{k \neq k_u^*} \Delta_{uk} \mathbb{P}\left[\mathbb{1}\{\kappa_u(t) = k\} = 1\right]$$

$$\geq \lambda_{min} \Delta \sum_{k \neq k_u^*} \mathbb{P}\left[\mathbb{1}\{\kappa_u(t) = k\} = 1\right]. \tag{32}$$

Therefore,

$$\sum_{s=1}^{t} \sum_{u \in [U]} \Psi_u(s) \geq \lambda_{min} \Delta \sum_{u \in [U]} \sum_{k \neq k_u^*} \mathbb{E}\left[T_{uk}(t)\right].$$

We now claim that

$$\sum_{u \in [U]} \Psi_u(t) \geq \frac{U(K-1)}{8t} \lambda_{min} D(\boldsymbol{\mu})(1-\alpha) \tag{33}$$

for infinitely many $t$. This follows from part (a) of Corollary 20 and the following fact:

**Fact 1.** *For any bounded sequence $\{a_n\}$, if there exist constants $C$ and $n_0$ such that $\sum_{m=1}^{n} a_m \geq C \log n \ \forall n \geq n_0$, then $a_n \geq \frac{C}{2n}$ infinitely often.*

Similarly, for any $u \in U$, it follows from parts (b) and (c) of Corollary 20 that

$$\Psi_u(t) \geq \frac{\max\{U-1, 2(K-U)\}}{8t} \lambda_{min} D(\boldsymbol{\mu})(1-\alpha) \tag{34}$$

for infinitely many $t$. $\qquad\square$

### 8.2.2 Early Stage: Proof of Theorem 17

In order to prove Theorem 17, we first derive, in the following lemma, a lower bound on the queue-regret in terms of the expected number of sub-optimal schedules.

**Lemma 22.** *For any system with parameters $(\boldsymbol{\lambda}, \boldsymbol{\mu})$, any policy, and any user $u \in [U]$, the regret is lower bounded by*

$$\Psi_u(t) \geq \sum_{k \neq k^*} \Delta_{uk} \mathbb{E}\left[T_{uk}(t)\right] - \epsilon_u t.$$

*Proof.* Since $Q_u(0) \sim \pi_{\lambda_u, \mu_u^*}$, we have,

$$
\begin{aligned}
\Psi_u(t) &= \mathbb{E}\left[Q_u(t) - Q_u^*(t)\right] \\
&= \mathbb{E}\left[Q_u(t) - Q_u(0)\right] \\
&\geq \mathbb{E}\left[\sum_{l=1}^{t} A_u(l) - S_u(l)\right] \\
&= \lambda_u t - \sum_{k=1}^{K} \mathbb{E}\left[T_{uk}(t)\right] \mu_{uk} \\
&= \lambda_u t - \left(t - \sum_{k \neq k_u^*} \mathbb{E}\left[T_{uk}(t)\right]\right) \mu *_u - \sum_{k \neq k_u^*} \mathbb{E}\left[T_{uk}(t)\right] \mu_{uk} \\
&= \sum_{k \neq k_u^*} \Delta_{uk} \mathbb{E}\left[T_{uk}(t)\right] - \epsilon_u t.
\end{aligned}
$$

$\qquad\square$

We now use this lower bound along with the lower bound on the expected number of sub-optimal schedules for $\alpha$-consistent policies (Corollary 20).

*Proof of Theorem 17.* To prove part (a) of the theorem, we use Lemma 22 and part (a) of corollary 20 as follows: For any $\gamma > \frac{1}{1-\alpha}$, there exist constants $C_4$ and $\tau$ such that for all $t \in$

Figure 3: Comparison of queue-regret performance of Q-ThS, Q-UCB, UCB-1 and Thompson Sampling in a 5 server system with $\epsilon_u = 0.15$ and $\Delta = 0.17$. Two variants of Q-ThS are presented, with different exploration probabilities; note that $3K \log^2 t/t$ is the exploration probability suggested by theoretical analysis (which is necessarily conservative). Tuning the constant significantly improves performance of Q-ThS relative to Thompson sampling.

$$[\max\{C_4 K^\gamma, \tau\}, (K-1)\frac{D(\boldsymbol{\mu})}{4\bar{\epsilon}}],$$

$$\frac{1}{U} \sum_{u \in [U]} \Psi_u(t) \geq \frac{\Delta}{U} \sum_{u \in [U]} \left( \sum_{k \neq k_u^*} \mathbb{E}\left[T_k(t)\right] - \epsilon_u t \right)$$

$$\geq (K-1)\frac{D(\boldsymbol{\mu})}{2} \left((1-\alpha)\log t - \log(KC_4)\right) - \bar{\epsilon} t$$

$$\geq (K-1)\frac{D(\boldsymbol{\mu})}{2} \frac{\log t}{\log \log t} - \bar{\epsilon} t$$

$$\geq (K-1)\frac{D(\boldsymbol{\mu})}{4} \frac{\log t}{\log \log t},$$

where the last two inequalities follow since $t \geq C_4 K^\gamma$ and $t \leq (K-1)\frac{D(\boldsymbol{\mu})}{4\bar{\epsilon}}$.

Part (b) of the theorem can be similarly shown using parts (b) and (c) of corollary 20. $\qquad\square$

**Additional Discussion:** As mentioned in Section 7, we note that (unstructured) Thompson sampling [20] is an intriguing candidate for future study.

In Figure 3, we benchmark the performance of Q-ThS against unstructured versions of UCB-1, Thompson Sampling and also a structured version of UCB (Q-UCB) analogous to Q-ThS. Note that there are two variants of Q-ThS displayed: the first has exploration probability $3K \log^2 t/t$, as suggested by the theory; the second has a tuned constant, with an exploration probability of $0.4K \log^2 t/t$.

It can be observed that in the early stage the unstructured algorithms perform better which is an artifact of the extra exploration required by Q-UCB and Q-ThS. In the late stage we observe that Q-UCB gives marginally better performance than UCB-1, however Thompson sampling has the best performance in both stages. This opens up additional research questions, discussed in Section 7. Q-ThS is dominated as well, but can be made to nearly match Thompson sampling by tuning the exploration probability (cf. the discussion above).

Nevertheless, it appears that Thompson sampling dominates UCB-1, Q-UCB, and the theoretically analyzed version of Q-ThS, at least over the finite time intervals considered. In some sense this is not surprising; empirically, similar observations in standard bandit problems [24, 25] are what have led to a surge of interest in Thompson sampling in the first place. Given these numerical experiments, it is important to quantify whether theoretical regret bounds can be established for Thompson sampling (e.g., in the spirit of the analysis in [26, 6, 27]).