[Reviews · NeurIPS 2016]

Reviewer 1

Summary

The paper considers the problem of assigning servers to serve a single queue with the aim of minimizing the queue length. In each time step, a server serves a request with a certain probability, and the goal of the paper is to minimize queuing regret: the expected excess queue length compared to the optimal policy that always uses the best server (the one with the highest service probability). More generally, in the appendix, the case of multiple queues and servers are considered. Asymptotic lower bounds are given on the queuing regret, as well as a modified version of Thompson sampling that achieves the optimal 1/t rate up to logarithmic factors.

Qualitative Assessment

Added after the rebuttal: - How surprising the results are: Clearly, the first intuition you provide is very naive, as the number of steady state time slots in a stable queuing system grows linearly with time, which is enough to compensate for the exploration (as it is evident from the results, too). Thus, claiming that the results are very surprising based on a wrong intuition is very misleading (several reviewers noted that the results are not surprising). - Motivation: please include the specific example you provided in the paper. - General: While I like the problem formulation, I think the solution should be much more general: Intuitively, any bandit algorithm with sublinear regret should achieve only a constant increase in the queue length (as follows from my original intuitive explanation). It would be great if the authors could generalize their results towards this direction, and would improve the paper a lot. This would also address the concerns raised by other reviewers that the paper reproves several known result from the literature, which would not be necessary if general reduction techniques were applied (perhaps it could also improve the constants I was complaining about). I also think that based on the experiments presented in the paper, it is a stretch to claim that one algorithm is better than the other. ======================================================== While the problem considered seems to be technically interesting, I have problems of seeing its motivation. Indeed, several practical systems employ queuing, but in those applications jobs are usually sent to servers where they queue (they do not wait in a central queue), and a main bottleneck is to observe the queue length at different servers (see the literature on load balancing, e.g., Ying, Srikant and Kang, "The Power of Slightly More than One Sample in Randomized Load Balancing," INFOCOM 2015). On the technical side, the results are not at all surprising (albeit it seems the authors are trying to sell the results as unexpected): In fact, running any usual bandit algorithm would result in selecting suboptimal servers O(log t) times, resulting in an average service rate of roughly (1 - Delta log t/t) mu^*, which naturally implies an expected queue length very close to the optimal with service rate mu^* if both service rates are larger than the arrival rate. I would really like to see a discussion about this. The proof technique is to look at the recurrence time of the queue induced by the policy (i.e., how often it returns to zero). By stability, this time is not too long with high probability, while the regret accumulated between short periods is not too large. While the analysis is given for a special algorithm, the results hinge upon a some very specific properties of the algorithm that seem to be easily satisfied by other bandit methods (e.g., UCB with slightly increased confidence intervals, see, Audibert, Munos, and Szepesvari, "Exploration-exploitation trade-off using variance estimates in multi-armed bandits," Theoretical Computer Science, 410:1876-1902, 2009). If I am correct, it would be much preferred to see such a treatment. The presentation is good in general, however, some symbols are not defined (or I failed to find their definitions, such as K, lambda (and its connection to A), Delta, connection of mu and S. I also had problems of understanding perfectly the model before its formal definition. Adding some explanations as suggested above and toning down the plausibility of the incorrect intuition (which lacks any queuing-related intuition) would improve the paper, in my opinion. Some additional technical questions: - Why is Assumption 1 needed (it would be easy to incorporate the starting queue length using standard queuing analysis techniques about convergence)? - Why don't you have the standard KL term of Lai and Robbins in the bound of Theorem 1? - It would be good to see some discussion about the several parameters of Corollary 5, in particular, its effect about switching limits in t and epsilon. - Were the experiments repeated? It would be good to see some error bars. I would also expect that increasing the confidence intervals in UCB has similar effects as increasing the exploration rate of Q-ThS (as it increases exploration). Minor comments: - Correct the wording of Definition 1. - line 302: correct "does does" - Section 6: mention that the missing simulation results are in the appendix.

Confidence in this Review

2-Confident (read it all; understood it all reasonably well)


Reviewer 2

Summary

The paper considers a variant of the multi-armed bandit problem in which it may take several slots to obtain a reward, and so arriving jobs accummulate in queues. Such problems have been heavily studied in the "queueing theory" field, where long-term performance is analyzed assuming that all the parameters are known. In this paper the authors assume that the "service rate" (i.e. the time it on average takes to obtain the reward) is unknown and needs to be learnt. The paper defines a queue-regret and analyzes its behavior. It establishes an asymptotic lower bound and designs an algorithm that achieves this bound up to a poly log factor. The analysis is non-trivial, as there are two phases: the queues first possibly build up until sufficient learning took place to stabilize them.

Qualitative Assessment

While the analytical results are impressive, the paper is not easy to read and confuses the reader. The appendix includes theorems and proofs for a multi-queue setting, while the main body is devoted to a single-queue setting. The introduction and abstract mix both and do not convey properly what the paper is about. Below are my concerns and some suggestions. 1. There is a lot of confusion w.r.t. queueing terminology and setting. I applaud the authors to step outside of their comfort zone, but the paper would hugely benefit from getting more familiar with queueing theory. It is essential to properly explain in the introduction the queueing problem you consider, e.g. do you consider a discrete or continuous time, how many servers you consider, are these pre-emptive, do they serve in parallel, how many queues are there, are the jobs routed immediately after their arrival (to a queue) or when a server becomes idle (to a server), are all the arriving jobs of the same class? In fact, there are a lot of queueing settings, and your single-queue model I believe best fits the problem "routing to parallel servers", which you however not even mention. You consideration of multiple queues is often described as "multi-class" (there are multiple classes of jobs that arrive, each class has a particular set of parameters - in your case \lambda's and \mu's). Other problems, e.g. scheduling, are related but fundamentally different. This will help your paper to have a larger impact as bridging the gap between machine learning and queueing theory. 2. Although catchy, your title is not specific enough. It is not clear what "queueing" means - as there are many queueing settings one could consider. It is not clear what "bandits" means - bandit feedback, bandit algorithm (which?), bandit model (which?). 3. There has been recent research on formulating queueing systems using bandit models. These are of at least 3 kinds: (1) job scheduling problems (see Gittins book 1989, and several papers by Ayesta and his co-authors since 2009), where each server selects which job to serve; (2) congestion control problems (e.g. Jacko & Sanso 2012 and Avrachenkov et al 2013), where users sending TCP packets are trying to learn the optimal transimission rate the Internet allows; (3) routing problems (see several papers by Glazebrook, Nino-Mora and their co-authors) where each job select which server to join. This literature may help you clarify which queueing system you consider in your paper (and perhaps give ideas for future research). 4. There has been recent research on queueing systems with unknown service rate, which you have completely ignored. It is essential that you review those papers and clarify what has already been done and what is different in your paper. E.g., Guo and Matta 2002, Zikos and Karatza 2011, Debo, Veeraraghavan 2014 and the papers citing these. 5. The abstract need to make clear what problem you consider. Is mentioning of the bandit model really essential there? I also don't think the sentence starting "A naive view of this problem would suggest..." is appropriate for an abstract. 6. I would encourage you to describe in the introduction the full problem (with multiple queues). This is because you should also refer to the related literature of the full problem. After you describe the queueing problem in the introduction, make very clear how the bandit model is used to model it: an "arm" could be a single job, or a queue of jobs, or a server (or many other things); how many arms are pulled every time; what happens if there is no job to serve; what what happens if the job is not completed (no reward), what are the possible rewards. In Section 3 you can then limit to case U=1 if you wish to present this problem only. 7. Your Assumption 1 (Initial state) seems very strong to me. In practice it will almost never be satisfied. There needs to be a discussion on whether it is essential for your results and what happens when it is not satisfied. 8. Several references have typos in the titles (e.g. kl-ucb, thompson, c|mu, bayesian) 9. Several unclear phrases are listed below - line 42: "incorporating queueing behavior into the MAB model is an essential challenge" -> it sounds like it has not been done before, but see the literature listed above - line 48: "reward equals job service" -> "reward is 1 of job service is completed in the current slot, 0 otherwise." - line 50 is correct only in the single server setting - line 57: "formally" - make it more formal (are you interested in finite-time or asymptotic results?) - line 62-64: this is far from obvious, mostly because the model is not clear yet - line 66: "unable to even stabilize" - this will depend on the parameters (for some it may be always stable) - line 77: "below zero" -> why not "negative"? - line 105: "up to a log log t factor" -> maybe more precisely "up to a 1 / log log t factor" - line 148 (and elsewhere): "arrival rate" and "service rate" are appropriate in a continuous time setting. Here it would be more precise to use "arrival probability" and "service completion probability" - line 154 (and in the appendix): "service offered" - define... what are the possible values? - equation above line 157 is repeated (1) - line 195: Defition 1 is wrongly worded. It is also not clear whether being alpha-consistent is problem-instance dependent - line 205: "bandit algorithms" - which of them do you mean? - line 213 (and lines 261, 284): give a link where the full proof can be found - line 225: it is not clear if you start from t=0 or t=1 - line 238-240: repertitive to 187-189 - line 267: "the the" - line 316: you mention comparison, without linking to where it can be found - "stability region" - define - "crowdsourcing example described in the introduction" - I haven't found one - line 436: the sentence is not finished. - Theorem 16 is different from Theorem 1. It is not obvious how Theorem 1 is obtained, nor it is mentioned that the proof of Theorem 16 is also the proof of Theorem 1. - notation \mu* and \mu_max is used inconsistently - Figure 3 and "Additional discussion" should be moved to the main body if space permits - Please add arrows in Figure 3 to indicate which curve refers to which policy, as it is not clear (I only see 4 curves)

Confidence in this Review

3-Expert (read the paper in detail, know the area, quite certain of my opinion)


Reviewer 3

Summary

The paper proposes to study a problem of allocating sequentially a queue of stochastically arriving jobs to servers. The problem is to try to use as much as possible the best (fastest) servers to run the jobs. It is therefore similar to a classical stochastic bandits problem except that using a suboptimal servers will make the queue increases while choosing the optimal server will make the queue decrease. The analysis shows that the size of the queue will grow logarithmically until we are able to identify the best arm, after which the queue will decrease with a rate of 1/t. The proposed algorithm is a version of Thompson Sampling with an extra forced uniform exploration.

Qualitative Assessment

The proposed problem is a simple and first step to combine bandit with queue problems. Thought the results are not very surprising the theoretical work seems to be very careful. The introduction is not very clear to me so its hard to follow the discussion on late stage/early stage before the formal definition of the problem. Maybe it could be nice to have the formal definition of the problem earlier. You say, line 136: 'Since these policies deal with optimality with respect to infinite horizon costs, ... they give steady-state and not finite-time guarantees.' I am not sure if I understand what you mean here. Can't you follow the type objective and analysis that is in the following paper Regret Bounds for Restless Markov Bandits by Ronald Ortner et.al. There, finite sample bounds on the regret are derived with an objective that compares the cumulative reward of the learner to some optimal policy (not a fixed arm strategy). But would it make sense to have a formulation of your problem that involves MDP and an optimal policy instead of comparing to the best fixed arm strategy? In the problem formulation it seems you don't make clear what the learner observations are. I mean you should add a sentence saying that R_k(t) is sampled from a distribution with mean \mu_k and that the learner only observes S(t) and no the other ones. Am I right? -in theorem 2, a function v(t) appears in the theorem but it is not clear what is the behaviour of this function. Please be more clear. -why does the quantity involved in the theorem 4 and 1 , D(mu) involves mu* and mu_min instead of the more traditional minimum gap or sum of the inverse squared gaps that appear in traditional bandits? - the experiments (Figure 3) seems to show that the extra exploration is not needed in practice. Do you think it can be possible to prove the same results that the one you have for the original thompson sampling? minor: - it seems that the notation is twice the same at line 187 and at line 238 - line 302: 'it does does not' - Figure 3 (in Appendix) : 'amd' -> and

Confidence in this Review

2-Confident (read it all; understood it all reasonably well)


Reviewer 4

Summary

The paper considers a variant of multiarmed bandit problem under queuing setting. In this problem, it is assume that there is only one queue and multiple servers. Arrivals to the queue and service offered by the servers are i.i.d Bernouli distributions across time. Any time that a job arrives it must be assigned to one of the servers with unknown statistical distributions (one can view the selected server as the arm which is pulled at that time instant in the MAB). At any discrete time, the queue can be served by at most one server and the problem is to schedule a server in every time slot. The goal of the paper is to see how queueing behavior impacts regret minimization in bandit algorithms, i.e., the expected queue length under a bandit algorithm with the corresponding one under a genie policy that always chooses the arm with the highest expected reward. The challenge of establishing the result in this paper is because the tradeoff between exploration-exploitation under queuing setting is more pronounced than the standard MAB problem, due to the fact that one must also consider stabilizing the queue while designing effective learning algorithm to minimize the regret. This results to a phase transition in the behavior of the regret, namely early stage and late stage. In the early stage, before the queue is stabilized, the regret grows poly-logarithmically similar to the classical MAB setting. But when the algorithm has learned the parameters, the system is stabilized, and the queue-length goes through regenerative cycles as the queue become empty. In particular, the paper give a lower bound in the switching time from the early stage to the late stage scales in terms of the number of servers and the gap between the arrival rate and the service rate of the optimal server. Moreover, it is shown that the early stage of the queueing bandit eventually goes to the late stage, in which the optimal queue-regret is at most O(1/t). In addition, the paper provides an algorithm based on Thompson sampling for Bernoulli arms and shows that it is nearly optimal in the time it takes to switch from the early stage to the late stage. Finally, the derived bounds and their dependency on the parameters of the problem are justified using some simulation results.

Qualitative Assessment

The notation part is repeated twice in the paper: lines 238 and 187 Please remove the existing proofs from the supplementary parts, e.g., Lemma 18. Moreover, it turns out that some of the proofs can be written in a shorter way by referring to some of the existing results from MAB problem and queuing systems, as shortening the proofs can make the paper sharper and easier to be followed. There are a few typos: e.g., lines 268, or 302. The valid range of parameters in which each of the main theorems hold are a bit confusing. This is because the statements have many parameters, e.g. Theorem 2. Maybe, it would be also more effective to illustrate these bounds pictorially as what has been done in Fig 1. In definition 1, when \alpha is very close to 0, this roughly means that an \alpha-consistent policy schedules almost all the jobs to the optimal server, which means that the regret of such policy must be very small. However, in Theorem 1, the lower bound on the regret will even increase as \alpha approaches zero, which is opposite to the former. Although it does not mean that the bound is not correct but it is somehow counter-intuitive. Is there any reason why this happens or am I missing something? Please bring the exact definition of Q-ThS from the supplementary file into the main manuscript.

Confidence in this Review

2-Confident (read it all; understood it all reasonably well)


Reviewer 5

Summary

The authors study a variant of the multi-armed bandit problem in which the arms are servers with unknown service rates, and jobs must queue for service. A natural analogue notion of "queue-regret" is introduced and analyzed. The authors conclude that the regret has an "early stage" in which the behavior is logarithmic (as in the usual MAB problems) but it's asymptotic behavior is in fact O(1/t) and study this threshold behavior.

Qualitative Assessment

Studying non-asymptotic behavior of such algorithms is in general quite interesting, and points for novelty were given due to this fact. I am less convinced of the practical motivation of the problem, although the problem as stated is not new. In particular, the fact that the queue can be served by at most one server in any given time step seems somewhat odd to me, but perhaps this captures more general settings. I am not familiar with queuing theory, and some discussion on this would be appreciated. Overall, I found the paper relatively easy to read. Effort was put into the presentation of the results (e.g., Figure 1), which was nice as a reviewer. There are some small inconsistencies (e.g., definitions starting in line 238 already appeared above). I also think it would be important to include Figure 3 in the main body of the paper, as I was certainly wondering how other algorithms compared. On a technical side, many pieces of the proofs follow or can be significantly shortened using known results from the MAB literature and algorithms. I appreciate the completeness, but it would be more readable to simply point out where things follow so the relevant parts can be skipped by a knowledgeable reader. More generally, it seems to me that a similar treatment using, e.g., UCB (with the appropriate confidence intervals) would also work -- is that so?

Confidence in this Review

2-Confident (read it all; understood it all reasonably well)


Reviewer 6

Summary

The authors propose a novel model-free approach for solving a MAB formulation of a job scheduling problem. The algorithm leverages a RL approach for exploring allocation amongst K servers in order to minimize regret, which they define as the difference between arriving (queued) jobs and the rate of compeleted jobs. They provide theoretical bounds on the diminishing queue-regret and demonstrate its effectiveness on a simulated problem domain.

Qualitative Assessment

This paper is quite theoretically strong and NIPS is an excellent venue for exploring the topics the authors tackle. The model-free nature of the algorithm foregoes many of the difficulties associated with applying scheduling algorithms in real-world settings, and the authors do a good job of applying contemporary MAB approaches to the topic. I am surprised at the lack of exploration of traditional scheduling algorithms, with the focus spent on solutions to MAB problems, though they would require significant alteration to adhere to the model-free nature of the domain and potentially come without any sort of statistical guarantee. That being said, I have two outstanding concerns about the clarity of the work. First, it is unclear whether the length of job completion is a metric in the algorithm, which doesn't appear in the body of the work. I am not certain how the algorithm handles jobs that take longer than a time step to complete. Presumably the same server continues executing the job in subsequent time steps, with a completion rate of zero until the job exits the system. Otherwise, does the job fail and remain in the queue until it has been allocated to a server a sufficient number of times? If so, I wonder what sort of improvements may result from relaxing the model-free nature of the algorithm to generate an expectation over job length and allocate accordingly. My only concern here is that job length is initially discussed in the introduction but isn't explained satisfactorily in the following sections. A second minor concern is the clarity of the problem domain section. I think it would help significantly to have a paragraph illustrating a problem domain (such as a ride share program extensively mentioned in the introduction), with more concrete description of the dynamics of the system as it relates to the algorithm. The theory is very strong in this work, but was quite difficult to penetrate when applying it to a concrete setting. As a first examination of applying MAB to scheduling in relation to queue-regret minization, this work provides a strong formulation with significant theoretical implications and, thus, merits publication.

Confidence in this Review

3-Expert (read the paper in detail, know the area, quite certain of my opinion)